# I-Spin live, an open-source software based on blind-source separation for real-time decoding of motor unit activity in humans

**Julien Rossato[1], François Hug[2,3], Kylie Tucker[3], Ciara Gibbs[4], Lilian Lacourpaille[1], Dario Farina[4], Simon Avrillon[4]***

[1]Nantes Université, Laboratory "Movement, Interactions, Performance" (UR 4334), Nantes, France; [2]Université Côte d'Azur, LAMHESS, Nice, France; [3]The University of Queensland, School of Biomedical Sciences, Brisbane, Australia; [4]Department of Bioengineering, Faculty of Engineering, Imperial College London, London, United Kingdom

**\*For correspondence:**
s.avrillon@imperial.ac.uk

**Abstract** Decoding the activity of individual neural cells during natural behaviours allows neuroscientists to study how the nervous system generates and controls movements. Contrary to other neural cells, the activity of spinal motor neurons can be determined non-invasively (or minimally invasively) from the decomposition of electromyographic (EMG) signals into motor unit firing activities. For some interfacing and neuro-feedback investigations, EMG decomposition needs to be performed in real time. Here, we introduce an open-source software that performs real-time decoding of motor neurons using a blind-source separation approach for multichannel EMG signal processing. Separation vectors (motor unit filters) are optimised for each motor unit from baseline contractions and then re-applied in real time during test contractions. In this way, the firing activity of multiple motor neurons can be provided through different forms of visual feedback. We provide a complete framework with guidelines and examples of recordings to guide researchers who aim to study movement control at the motor neuron level. We first validated the software with synthetic EMG signals generated during a range of isometric contraction patterns. We then tested the software on data collected using either surface or intramuscular electrode arrays from five lower limb muscles (gastrocnemius lateralis and medialis, vastus lateralis and medialis, and tibialis anterior). We assessed how the muscle or variation of contraction intensity between the baseline contraction and the test contraction impacted the accuracy of the real-time decomposition. This open-source software provides a set of tools for neuroscientists to design experimental paradigms where participants can receive real-time feedback on the output of the spinal cord circuits.

## eLife assessment

This article compiles existing algorithms into an open-source software package that enables real-time (and offline) motor unit decomposition from muscle activity collected via grids of surface electrodes and indwelling electrode arrays. The package is **valuable** given that many motor neuroscience labs are using such algorithms and that there exists a host of potential applications for such data. Validation of the software package is **compelling**, suggesting that it can be successfully applied across a range of muscles and tasks.

## Introduction

Motor units transduce the neural signals that originate from supraspinal centres, spinal circuits, and sensory systems into force (*Sherrington, 1925*). In healthy individuals, each action potential propagating along the axon of an alpha motor neuron elicits action potentials in all its innervated muscle fibres. The activity of motor neuron – in the form of spike trains – represents the neural code of movement to muscles. Decoding this firing activity in real time during various behaviours can thus substantially enhance our understanding of movement control (*Basmajian, 1963*; *Formento et al., 2021*; *Bräcklein et al., 2022*; *Rossato et al., 2024*). Real-time decoding is also essential for interfacing with external devices (*Farina et al., 2023*) or virtual limbs (*Oliveira et al., 2024*) when activity is present at the periphery of the nervous system. For example, individuals with a spinal cord injury can control a virtual hand with the residual firing activity of the motor units in their forearm (*Oliveira et al., 2024*). Furthermore, sampling the activity of motor units receiving a substantial portion of independent synaptic inputs may pave the way for movement augmentation – specifically, extending a person's movement repertoire through the increase of controllable degrees of freedom (*Eden et al., 2022*). In this way, *Formento et al., 2021* showed that individuals can intuitively learn to independently control motor units within the same muscle using visual cues. Having access to open-source tools that perform the real-time decoding of motor units would allow an increasing number of researchers to improve and expand the range of these applications.

To date, researchers classically identify a few motor units during relatively weak contractions using concentric needle or fine wire recording electrodes (*LeFever and De Luca, 1982*). They then separate the overlapping spike trains with a spike-sorting algorithm (e.g. *McGill et al., 2005*). Recent developments of intramuscular (*Farina et al., 2008b*; *Muceli et al., 2015*; *Muceli et al., 2022*; *Chung et al., 2023*) and surface (*Farina et al., 2016*; *Caillet et al., 2023*) electromyography (EMG) electrode arrays facilitate both a larger recording zone (i.e. the volume from which motor unit action potentials are recorded), and the recording of the same motor unit action potential across multiple channels. In conjunction with this hardware advance, the development of novel EMG decomposition software/programs, such as multichannel spike-sorting (*Rey et al., 2015*; *Buccino et al., 2020*; *Steinmetz et al., 2021*; *Pachitariu et al., 2023*) and blind-source separation (*Holobar and Zazula, 2007*; *Holobar and Farina, 2014*; *Farina and Holobar, 2016*; *Negro et al., 2016*; *Chen et al., 2019*) algorithms, enables a relatively large number of individual motor units to be decoded from each recording (*Muceli et al., 2015*; *Muceli et al., 2022*; *Chung et al., 2023*; *Caillet et al., 2023*). Contrary to spike-sorting algorithms, blind-source separation does not directly sort action potential shapes, but rather optimises a set of separation vectors, that is, motor unit filters, which maximises the sparseness of motor unit pulse trains from which discharge times are estimated (*Holobar and Zazula, 2007*; *Holobar and Farina, 2014*; *Farina and Holobar, 2016*; *Negro et al., 2016*; *Chen et al., 2019*). The current open-source implementations of this approach rely on offline processing, which restricts its ability to be used for neurofeedback and human interfacing technologies.

Recent studies have reported real-time capabilities of motor unit identification by adapting the offline blind-source separation algorithm (*Formento et al., 2021*; *Bräcklein et al., 2022*; *Barsakcioglu et al., 2021*; *Chen et al., 2020*; *Zheng and Hu, 2019*). These studies used a two-step approach: (1) the separation vector for each motor unit is identified with offline decomposition during the training phase and (2) the same vectors are applied in real time to new EMG recordings. The real-time identification of motor units is only used by a few specialised research teams, and there are no publicly available algorithms or user interfaces for this task. In addition, the accuracy and capabilities of online decomposition have not been systematically tested in multiple muscles or intensities.

Here, we provide an open-source software that can be used to visualise and track motor unit firing activities in real time. We document the software validity and its capabilities on data collected from five lower limb muscles (gastrocnemius lateralis and medialis, vastus lateralis and medialis, and tibialis anterior) during isometric contractions of varying force levels, using either surface or intramuscular electrode arrays. The real-time identification of motor unit firing activity was first validated using synthetic EMG signals. Then, the accuracy of the algorithm was determined during experiments from the rate of agreement calculated between the motor unit spike trains identified in real time and those identified offline after manual editing. Data, codes, and a user manual are available at https://github.com/simonavrillon/I-Spin (copy archived at *Avrillon, 2024*).

## Results

### Overview of the approach

An EMG signal represents the sum of trains of action potentials from all the active motor units within the recorded muscle volume (*Figure 1A*). During stationary conditions, for example, isometric contractions, the train of motor unit action potentials can be modelled as the convolution of series of discrete delta functions, representing the discharge times, and motor unit action potentials that have a consistent shape across time. When EMG signals are recorded with an array of electrodes, the shape of the recorded potential of each motor unit differs across electrodes. This is due to (1) the varying conduction velocity of action potentials among the muscle fibres and (2) the location/depth of the muscle fibres that belong to each motor unit relatively to the electrodes, which impact the low pass filtering effect of the tissue on the recorded potential. Increasing the number and density of recording electrodes increases the likelihood that each motor unit will have a unique motor unit action potential profile (shape), that is, a temporal and spatial profile that differs from all the other active motor unit within the recorded volume (*Caillet et al., 2023*; *Farina et al., 2008a*). The uniqueness of motor unit action potential profiles is necessary for the blind-source separation to accurately estimate the motor unit discharge times. Conversely, the spike trains of two motor units with similar action potential profiles will be merged by the model.

Our software uses a fast independent component analysis (fastICA) to retrieve motor unit spike trains from the EMG signals. For this, it iteratively optimises a separation vector (i.e. the motor unit filter) for each motor unit (*Figure 1B*; *Negro et al., 2016*; *Chen et al., 2019*; *Barsakcioglu et al., 2021*; *Negro et al., 2016*; *Chen et al., 2019*; *Barsakcioglu et al., 2021*). The projection of the EMG signals on this separation vector generates a sparse motor unit pulse train, with most of its samples close to zero and a smaller number of samples significantly greater than zero (*Figure 1B*). The discharge times are estimated from this motor unit pulse train using a peak detection function and a k-mean classification with two classes to separate the high peaks (spikes) from the low peaks (noise and other motor units). During the decomposition in real time, short segments of EMG signals are projected on the saved separation vectors, and the peaks are classified as discharge times if they are closer to the centroid of the class 'spikes' than to the centroid of the class 'noise' (*Figure 1C*). The algorithm used to identify motor units discharge activity is based on that proposed by *Negro et al., 2016* and *Barsakcioglu et al., 2021*.

### Overview of the software

The software used in this article was coded as a MATLAB app (version 2022a, The MathWorks, Inc, USA), but an alternative version coded with Python is also available. It allows researchers to record and process signals from surface and intramuscular electrode arrays using multiple acquisition systems (EMG-Quattrocento, OT Bioelettronica, Italy; Open Ephys acquisition board, Open Ephys, USA; Intan RHD recording system; Intan Technologies, USA). As the accuracy of the algorithm relies on the consistency of motor unit filters, it is recommended to record these EMG signals during stationary conditions – for example, isometric contractions – to limit changes in muscle geometry or position/orientation of the active muscle fibres relative to the electrodes (*Glaser and Holobar, 2019*; *Oliveira and Negro, 2021*). The framework to perform real-time identification of motor neuron activity has four steps (*Figure 1—figure supplement 1*). First, the EMG signal is recorded while participants perform a contraction at the requested intensity such that an *electrode mask* is manually generated to remove channels with artefacts or low signal-to-noise ratio. This *electrode mask* is then used for the rest of the experimental session. Second, the force offset is measured and removed before performing maximal voluntary contraction (MVC). The measured MVC is used to standardise all the submaximal isometric contractions. Third, a baseline contraction is performed at a force level close to, or slightly above, the intensity of the testing task, and the separation vectors are identified with offline blind-source separation. Fourth, the separation vectors are applied over incoming segments of EMG signals during a test contraction to identify motor unit firing activity in real time.

Three forms of feedback can be displayed to the participant: a raster plot of the discharge times for each motor unit of a given array, a quadrant displaying the firing rates of two motor units, and the smoothed firing rate of a given motor unit with a scrolling target to track (*Figure 1C*).

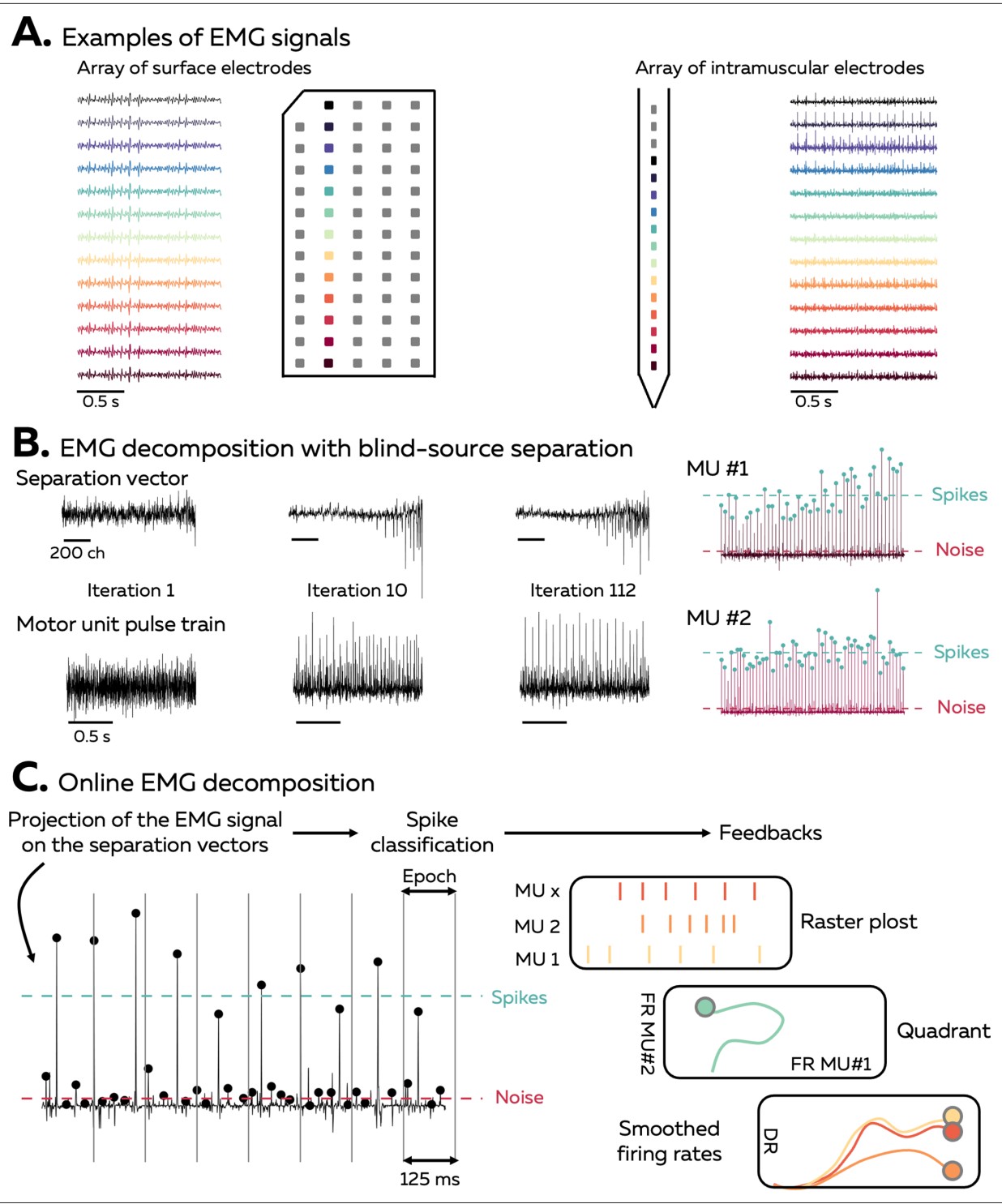

**Figure 1.** Overview of the approach. (**A**) During isometric contractions, electromyographic (EMG) signals can be considered as the sum of all action potentials that originate from the muscle fibres of all the active motor units that lie within the electrodes recording zone. The shape of the recorded action potentials differs across electrodes when recorded with an array of surface or intramuscular electrodes. The EMG signal and each individual MUAP profile depends on the position of the electrode, as highlighted by the different colours. (**B**) Decomposing EMG signals consists of solving the inverse problem, that is, to estimate the discharge times of the active motor units from the EMG signals. Our software uses a fast independent component analysis (fasICA) to optimise a set of separation vectors for each motor unit. To this end, each separation vector is iteratively optimised to maximise the sparseness of the motor unit pulse train. At the end of this step, the motor unit pulse train is refined, and a k-mean classification is applied to separate the high peaks, which represent the targeted motor unit spikes, from the low peaks (other motor units and noise). (**C**) During the online

*Figure 1 continued*

EMG decomposition, the extended EMG signals recorded over 125 ms segments are projected on the separation vectors, and the peaks are detected using the function 'islocalmax'. Each peak is classified as *spike* or *noise* depending on the distance separating them from the centroids of the classes identified during the calibration. At the end of this process, the motor unit firing activity is translated into visual feedback, in the form of a raster plot, a quadrant, or the smoothed firing rate of an identified motor unit.

The online version of this article includes the following figure supplement(s) for figure 1:

**Figure supplement 1.** Workflow of a typical experimental session with I-Spin live.

## Manual editing to improve the accuracy of the motor unit pulse train

Manual editing of motor unit discharge times identified with automatic decomposition (or spike-sorting) generally improves the accuracy of the spike trains (*Del Vecchio et al., 2020*; *Hug et al., 2021*; *Avrillon et al., 2024*). In the case of online decomposition, manual editing can be performed after the baseline contraction to improve the accuracy of the motor unit filters. It results in a motor unit pulse train with peaks easily separable from the noise. Several metrics have been proposed to automatically remove the unreliable motor unit pulse trains (*Negro et al., 2016*; *Holobar et al., 2014*). For example, the silhouette value (SIL) measures the normalised distance between the spikes and the noise after k-mean classification (*Negro et al., 2016*). As an example, *Figure 2A* displays the silhouette values of all the motor units identified after the baseline contraction in five different muscles (vastus lateralis [VL] and medialis [VM], gastrocnemius lateralis [GL] and medialis [GM], and tibialis anterior [TA]) with either grids of surface electrodes or intramuscular electrodes arrays. When considering the baseline contractions performed at 20% MVC, on average 20 ± 9 (VL), 14 ± 5 (VM), 25 ± 11 (GL), 19 ± 9 (GM), 15 ± 4 (TA grid), and 10 (TA intra) motor units per participant were identified. Their SIL values calculated before manual editing were 0.89 ± 0.05 for VL, 0.83 ± 0.04 for VM, 0.82 ± 0.03 for GL, 0.87 ± 0.04 for GM, 0.94 ± 0.01 for TAgrid, and 0.95 ± 0.02 for TAintra. After visual inspection, a significant number of motor units was removed as their pulse train showed no clear separation between the spikes and the noise. The number of motor units removed was 6 ± 5 for VL, 10 ± 5 for VM, 21 ± 12 for GL, 6 ± 5 for GM, and 1 for TA grid and TA intra (*Figure 2A*). The remaining motor units exhibited a SIL value of 0.91 ± 0.04, 0.89 ± 0.06, 0.89 ± 0.03, and 0.90 ± 0.02, 0.94 ± 0.01, and 0.95 ± 0.02 for VL, VM, GL, GM, TAgrid, and TA intra, respectively (*Figure 2B*).

When considering the baseline contraction performed at 40% MVC, on average 17 ± 6 (VL), 15 ± 5 (VM), 29 ± 13 (GL), and 13 ± 4 (GM) motor units were identified by automatic decomposition, with 12 ± 8 (VL), 3 ± 3 (VM), 5 ± 6 (GL), and 4 ± 3 (GM) motor units kept after visual inspection (*Figure 2A*). The SIL value of the selected motor units reached 0.91 ± 0.03 for VL, 0.89 ± 0.05 for VM, 0.87 ± 0.02 for GL, and 0.88 ± 0.03 for GM after manual editing (*Figure 2B*). These results show how manual editing can improve the accuracy of spike detection from the motor unit pulse trains. Moreover, a SIL value around 0.9 can be used as a threshold to automatically remove the motor unit pulse trains with a poor quality a priori. Thus, these two steps were performed in the all the subsequent analyses. Importantly, it is worth noting that the motor unit pulse train must always be visually inspected after the session to check for errors of the automatic identification of discharge times.

## Validation of the algorithm

We first validated the accuracy of the algorithm using synthetic EMG signals generated with an anatomical model entailing a cylindrical muscle volume with parallel fibres (see *Farina et al., 2008a*, *Konstantin et al., 2020* for a full description of the model). In this model, subcutaneous and skin layers separate the muscle from a grid of 65 surface electrodes (5 columns, 13 rows), while an intramuscular array of electrodes is directly inserted in the muscle under the grid with an angle of 30°. A total of 150 motor units were distributed within the cross-section of the muscle. Recruitment thresholds, firing rate/excitatory drive relations, and twitch parameters were assigned to each motor unit using the same procedure as *Fuglevand et al., 1993*. During each simulation, a proportional-integral-derivative controller adjusted the level of excitatory drive to minimise the error between a predefined target of force and the force generated by the active motor units.

*Figure 3A* displays the raster plots of the active motor units during simulated trapezoidal isometric contractions with plateaus of force set at 10, 20, and 30% MVC. A sinusoidal isometric contraction ranging between 15 and 25% MVC at a frequency of 0.5 Hz was also simulated. We identified on

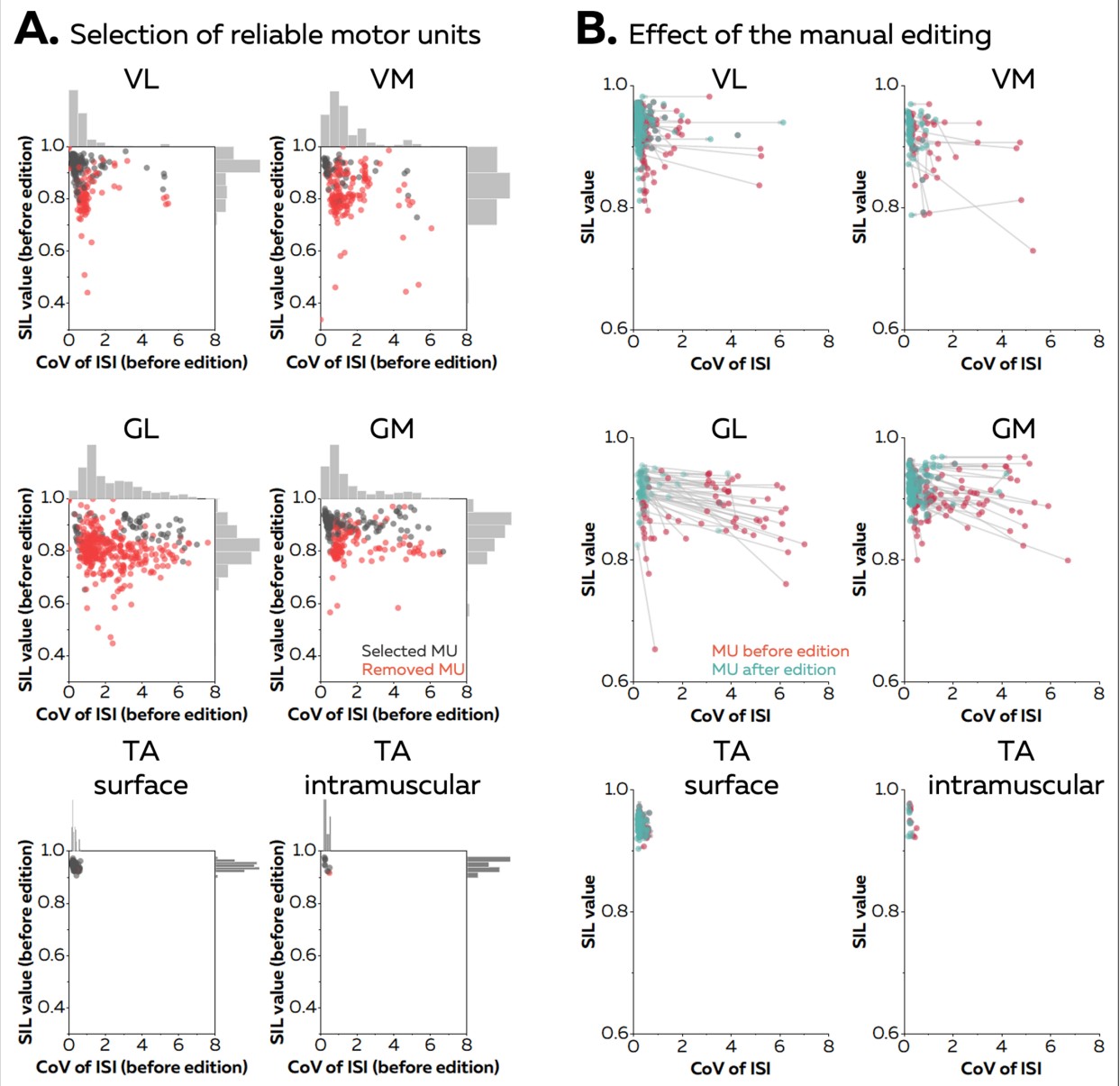

**Figure 2.** Effect of the manual editing on the reliability of motor unit pulse trains. Once the participants completed the baseline contraction, we ran an automatic offline decomposition. Then, an operator visually inspected and removed all the motor units for which the spikes were not clearly separated from the noise (red dots in **A**). The remaining motor units were manually edited (black dots in **A**). (**B**) The manual editing consisted of removing false positives and adding the false negatives, before updating the motor unit filter. The effect of this step on the SIL value and the coefficient of variation of the interspike intervals (CoV of ISI, without units) is shown on the right panel. The CoV of ISI estimates the regularity of spiking for each motor unit, an expected behaviour during isometric contractions at consistent levels of force. The red dots are the motor units before editing and the green dots are the motor units after editing. These scatters are connected with a grey vector to show the changes in SIL value and CoV of ISI. Vastus lateralis (VL), vastus medialis (VM), gastrocnemius lateralis (GL), gastrocnemius medialis (GM), and tibialis anterior (TA).

average 10 ± 1 and 12 ± 2 motor units with surface and intramuscular arrays, respectively (*Figure 3A*). During the offline decomposition, the rate of agreement between the identified discharge times and the ground truth, that is, the simulated discharge times, reached 100.0 ± 0.0% for intramuscular EMG signals and 99.2 ± 1.8% for surface EMG signals (*Figure 3B*). The offline estimation of motor unit filters was therefore highly accurate, independently of the level of force or the pattern of the isometric contraction.

Motor unit filters estimated during a baseline contraction at 20% MVC were then applied in real time on signals simulated during a contraction with a different pattern (sinusoidal; *Figure 3C*). The

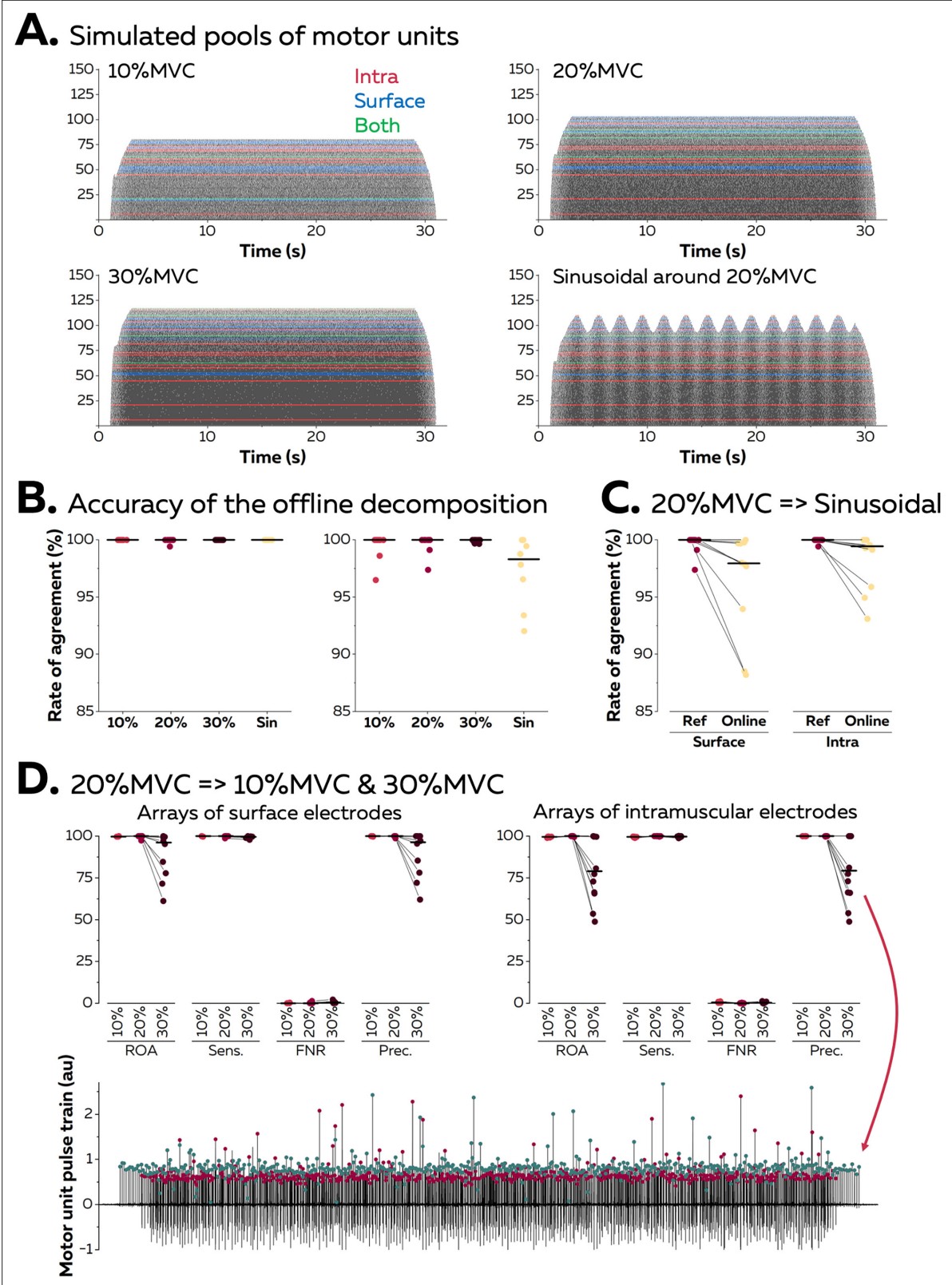

**Figure 3.** Validation of the algorithm with synthetic electromyographic (EMG) signals. Surface and intramuscular EMG signals were generated using an anatomical model with 150 motor units. (**A**) Raster plots of the active motor units during simulated trapezoidal contractions performed at 10, 20, and 30% of the maximal force (maximal voluntary contraction [MVC]), and during a sinusoidal contraction with the force varying between 15 and 25% MVC at a rate of 0.5 Hz. The spike trains in red, blue, or green were respectively identified from intramuscular, surface, or both EMG signals. (**B**) The identified

*Figure 3 continued on next page*

*Figure 3 continued*

discharge times were compared to the ground truth, that is, the simulated discharge times, using the rate of agreement. Each dot is a motor unit, and the line is the median. (**C**) The motor unit filters identified from the reference contraction (i.e. trapezoidal contraction at 20% MVC) were then applied in real-time on the incoming EMG signals simulated during the sinusoidal contraction. The rates of agreement are displayed for each motor unit; the line is the median. (**D**) The motor unit filters identified at 20% MVC were applied in real-time on signals simulated during contractions at 10 and 30% MVC. Rates of agreement, sensitivity (sens.), false negative rates (FNR.), and precision (prec.) are displayed for each motor unit. The lower precision for motor units identified in real time at 30% MVC can be explained by the presence of a *merged* motor unit, as highlighted on this example. The red dots represent the discharge times of this *merged* motor unit while the green dots represent the discharge times of the targeted motor unit.

rates of agreement between the online decomposition and the ground truth reached 96.3 ± 4.6% and 98.4 ± 2.3% for surface and intramuscular EMG signals, respectively. Finally, we tested whether the accuracy of the online decomposition changed when the level of force decreased or increased by 10% MVC when compared to the calibration performed at 20% MVC (*Figure 3D*). The rate of agreement remained high when applying the motor unit filters on signals recorded at 10% MVC: 99.8 ± 0.2% (surface EMG) and 99.5 ± 0.3% (intramuscular EMG). It is worth noting that only 3 out of 10 motor units identified from surface EMG at 20% MVC were active at 10% MVC, while 8 out of 12 motor units identified from intramuscular EMG were active at 10% MVC. This shows how the decomposition of EMG signals tends to identify the last recruited motor units, which often innervate a larger number of fibres than the early recruited motor units (*Henneman, 1957*). On the contrary, the application of motor unit filters on signals simulated at 30% MVC led to a decrease in the rate of agreement, with values of 88.6 ± 14.0% (surface EMG) and 80.3 ± 19.2% (intramuscular EMG). This decrease in accuracy did not impact all the motor units, with five motor units keeping a rate of agreement above 95% in both signals. For the other motor units, we observed a decrease in precision, which estimates the ratio of true discharge times over the total number of identified discharge times. This was caused by the recruitment of two motor units sharing a similar space within the muscle, which resulted in a merge in the same pulse train (*Figure 3D*).

## Application of motor unit filters in experimental data

We then asked eight participants (four males and four females) to perform trapezoidal isometric contractions with plateaus of force set at 10 and 20% MVC during which surface EMG signals were recorded from the TA with 256 electrodes separated by 4 mm. The aim of this experiment was to confirm the results of the simulation; specifically, to test the accuracy of the online decomposition when the level of force was below, equal to, or above the level of force produced during the baseline contraction used to estimate the motor unit filters (*Figure 4*). We assessed the accuracy of the motor unit spike trains identified in real time using their manually edited version as reference. A total of 144 motor units were identified at both 10 and 20% MVC. When the test signals were recorded at the same level of force as the baseline contraction, we obtained rates of agreement of 95.6 ± 6.8% (10% MVC) and 93.9 ± 5.9% (20% MVC). The sensitivity reached 95.9 ± 6.7% (10% MVC) and 94.4 ± 5.6% (20% MVC), and the precision reached 99.6 ± 1.3% (10% MVC) and 99.4 ± 1.9% (20% MVC).

When the filters identified at 20% MVC were applied on signals recorded at a lower level of force (10% MVC), the rates of agreement decreased to 87.9 ± 16.2%. The sensitivity also decreased to 88.0 ± 16.2%, but the precision remained high (99.4 ± 4.3). Thus, the decrease in accuracy was mostly caused by missed discharge times rather than the false identification of artefacts or spikes from other motor units. When the filters identified at 10% MVC were applied to signals recorded at a higher level of force, the rates of agreement decreased to 83.3 ± 13.5%. The sensitivity decreased to 90.7 ± 8.1%, and the precision also decreased to 90.9 ± 12.6%. This result confirms what was observed with synthetic EMG, that is, motor units recruited between 10 and 20% MVC can substantially disrupt the accuracy of the decomposition in real time, as highlighted in *Figure 4* (lower panel). Importantly, this situation does not happen for all the motor units, as suggested by the distribution of the values in *Figure 4*.

## Accuracy of the online decomposition in different muscles

Twenty-one male participants completed additional experiments to test the accuracy of the online decomposition in different muscles (vastus lateralis [VL] and medialis [VM], gastrocnemius lateralis [GL] and medialis [GM], and tibialis anterior [TA]). They all performed isometric contractions at 20 and 40%

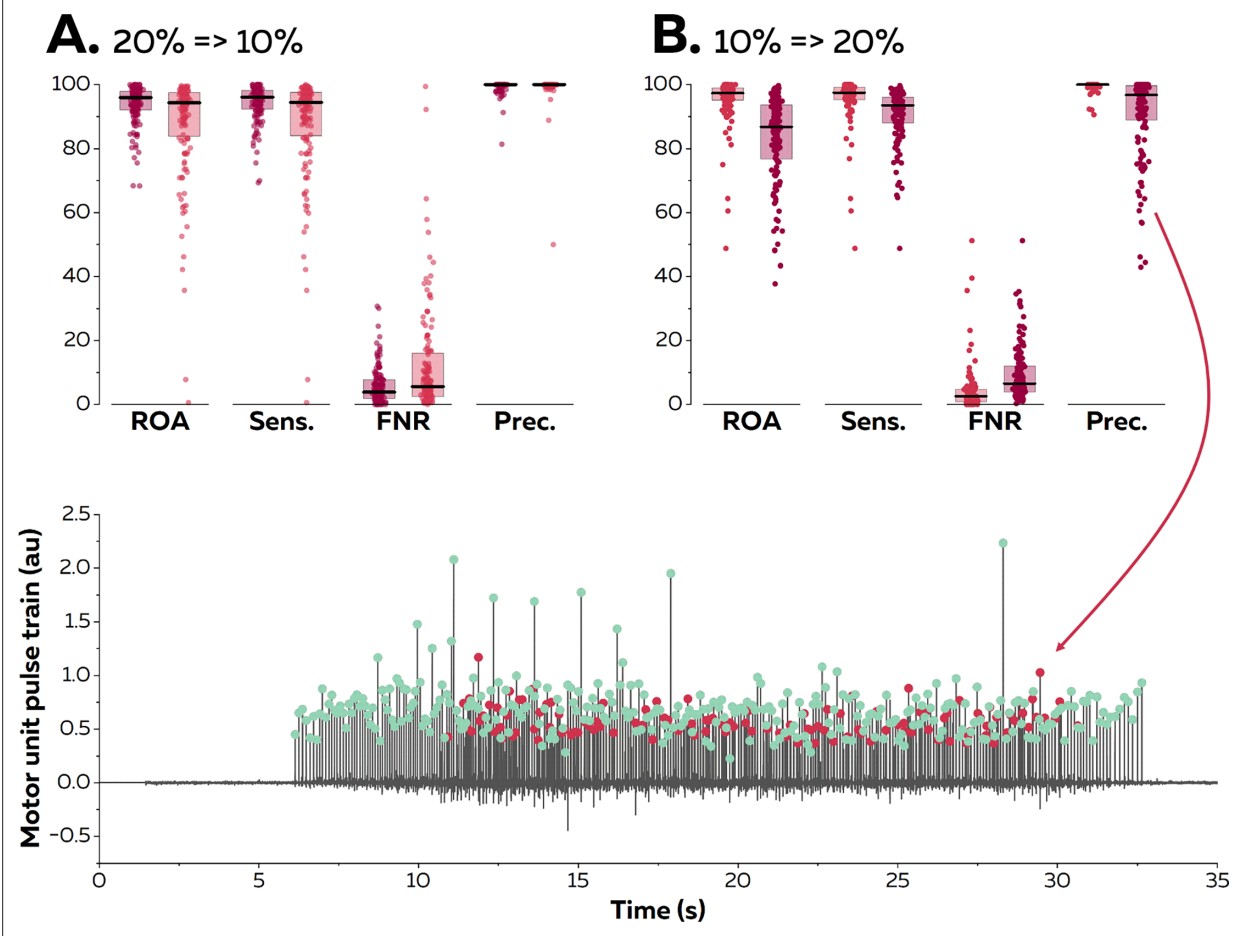

**Figure 4.** Reapplication of motor unit filters on electromyographic (EMG) signals recorded at varying contraction intensities. Surface EMG signals were recorded during isometric contractions at 10 and 20% maximal voluntary contraction (MVC). We compared the motor unit spike trains identified in real time with their manually edited version. The contraction intensity of the test contraction was either equal to (20% => 20% in **A**; dark red & 10% => 10% in **B**; light red), below (20% => 10% in **A**; light red), or above (10%=>20% in B; dark red) the level of the baseline contraction. We calculated the rate of agreement (ROA), the sensitivity (Sens.), the false negative rate (FNR.), and the precision (Prec.) for each motor unit. Each dot is an individual motor unit, each box represents the 25th and 75th percentiles of the distribution of values, and each line is the median. The lower precision in (**B**) was caused by the presence of a *merged* motor unit in the motor unit pulse train. The red dots represent the discharge times of this *merged* motor units while the green dots represent the discharge times of the targeted motor unit.

MVC while EMG signals were recorded with either grids of 64 surface electrodes (VL, VM, GL, GM, TA) or an intramuscular array of 16 electrodes (TA).

At 20% MVC, 94 (VL; eight participants), 21 (VM; eight participants), 14 (GL; eight participants), 56 (GM; eight participants), 68 (TA grid; five participants), and 9 motor units (TA intra; one participant) motor units were identified. At 40% MVC, 76 (VL), 21 (VM), 19 (GL), and 25 (GM) motor units were identified. For the sake of clarity, we only report in this section the rates of agreement for each intensity and muscle. Values of sensitivity, precision, and rates of false negatives are reported in *Figure 5*. The highest rate of agreement was observed for the TA (93.6 ± 9.2% with the grid and 97.3 ± 5.2% with the intramuscular array). Those values were lower for the vastii (VL: 82.1 ± 19.7%; VM: 75.3 ± 18.5%) and gastrocnemii muscles (GL: 88.1 ± 7.8%; GM: 81.0 ± 17.7). When considering the contractions performed at 40% MVC, the rate of agreement was 84.0 ± 15.6% for VL, 75.2 ± 20.6% for VM, 82.5 ± 8.1% for GL, and 87.9 ± 8.7% for GM (*Figure 5*).

## Accuracy of the real-time biofeedback

Because the accuracy of the raster plot feedback is directly related to the accuracy of the estimation of the discharge times reported in the previous sections, here we only focus on the accuracy of the

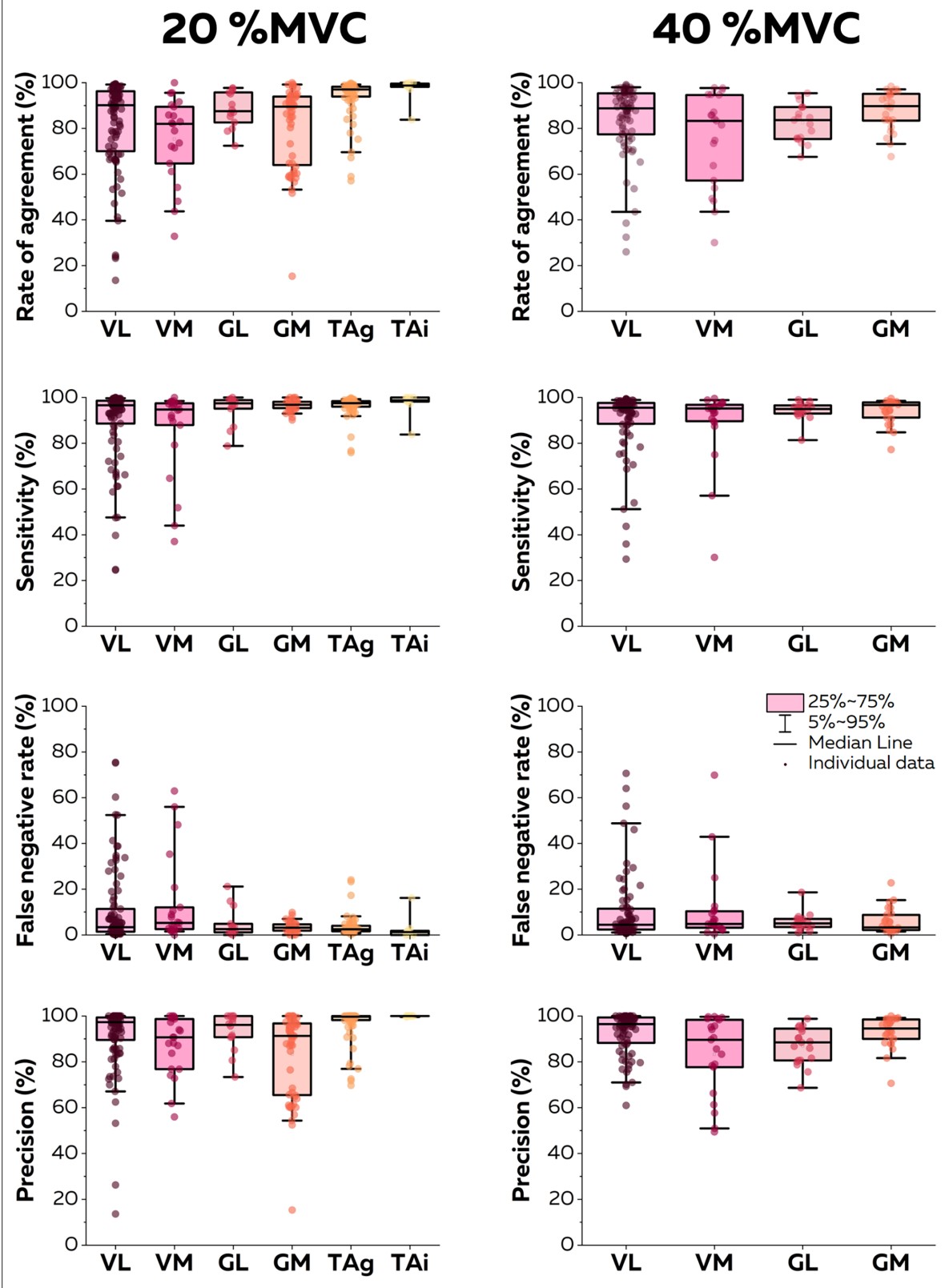

**Figure 5.** Accuracy of the online electromyographic (EMG) decomposition. We compared the motor unit spike trains identified in real time with their manually edited version. We calculated the rate of agreement, the sensitivity, the false negative rate, and the precision for each motor unit. Each dot is an individual motor unit, each box represents the 25th and 75th percentiles of the distribution of values, each bar represents the 5th and 95th percentiles of the distribution of values, and each line is the median. VL: vastus lateralis, VM: vastus medialis, GL: gastrocnemius lateralis, GM:

*Figure 5 continued*
gastrocnemius medialis, TAg: tibialis anterior recorded with a high-density grid of electrodes, and TAi: tibialis anterior recorded with an intramuscular array of electrodes.

smoothed firing rates. It can either be displayed in a quadrant, with a cursor moving within a two-dimensional space according to the firing rates of two motor units, or with one cursor for a given identified motor units moving in the vertical direction and following a scrolling path.

At 20% MVC, the root mean squared error (RMSE) between the firing rate estimated in real time and its edited version was consistently below two pulses per second for all the motor units (*Figure 6*). The lowest RMSE was observed with the TA muscle (0.64 ± 0.77 pps and 0.44 ± 0.43 pps with grids and intramuscular arrays of electrodes). RMSE was 1.4 ± 1.5 pps for VL, 1.8 ± 1.5 pps for VM, 1.7 ± 1.1 pps for GL, and 1.1 ± 1.1 pps for GM. At 40% MVC, the RMSE reached 1.1 ± 1.2 pps for VL, 1.8 ± 1.1 pps for VM, 0.8 ± 0.5 pps for GL, and 0.6 ± 0.4 pps for GM. Overall, these RMSE provides strong evidence that the biofeedback accurately reflects the motor unit firing activity. It is highlighted by the two examples displayed in *Figure 6B*, with a raw smoothed firing rate with an RMSE at 0.83 pps, and in *Figure 6C*, with firing rates displayed in a quadrant with an average RMSE of 1.75 pps. Videos of the different forms of feedback are available in the GitHub.

## Discussion
### Assumptions of the algorithm
There are processing assumptions for the blind-source separation algorithm to accurately identify motor unit discharge times from multichannel EMG signals. Among them, the most important is the uniqueness of the distribution of motor unit action potentials across electrodes (that defines the separation vector) among all the other active motor units within the recording volume (*Caillet et al., 2023*; *Farina et al., 2008a*). When this condition is not met, merged motor units appear in the motor unit pulse train, causing an increase in false positives (*Figures 3 and 4*) One way to satisfy this condition is to increase the selectivity of the electrodes to record the discharge activity of only a few motor units from a small volume of muscle (*LeFever and De Luca, 1982*). For example, *Figure 1A* shows motor unit action potentials detected only over 3–4 electrode sites along the array of intramuscular electrodes, while a motor unit action potential can be observed across many more electrodes with grids of surface electrodes. Therefore, the likelihood of having spatially overlapping motor unit action potentials – and thus merged motor units – is lower, which explains why the rate of agreement of motor units identified from intramuscular arrays of electrodes is much higher than grids of surface electrodes (*Muceli et al., 2015*; *Muceli et al., 2022*). A second way to increase the percentage of discriminable motor units among all the active motor units in the recording volume is to increase the spatial sampling of their activity using multiple electrodes (*Farina et al., 2016*; *Caillet et al., 2023*; *Farina and Holobar, 2016*; *Farina et al., 2008a*). This has led to the growth of EMG recording systems with dense grids of surface electrodes (*Farina et al., 2016*), which compensate for the lack of specificity of motor unit action potential profiles that are recorded when using a pair of traditional bipolar EMG electrodes (*Lindstrom and Magnusson, 1977*).

Another necessary condition for EMG decomposition using the blind-source separation algorithm is the consistency of the motor unit filters across time. An obvious reason inducing changes in motor unit filters would be the displacement of the electrodes relatively to the source. Such drifts also exist with intracortical arrays and can be corrected with appropriate methods that track waveforms across electrodes (*Steinmetz et al., 2021*). However, the geometry of the muscle tissue is much more variable than that of the cortical tissue, especially during movements. For example, muscle bellies become bulkier while shortening (*Herbert et al., 2019*), increasing the distance between the surface electrodes and the deep sources. In addition, the pennation angle of muscle fibres can change with contraction intensity (*Fukunaga et al., 1997*), modifying the direction of the propagation of motor unit action potentials along the fibres relatively to the position of the electrodes. All these factors impact the recorded motor unit action potential profiles across electrodes, which in turn will reduce the capacity to discriminate the same motor unit from the EMG signal (*Oliveira and Negro, 2021*; *Farina et al., 2004*; *Glaser et al., 2017*). For these reasons, we recommend applying our approach during isometric

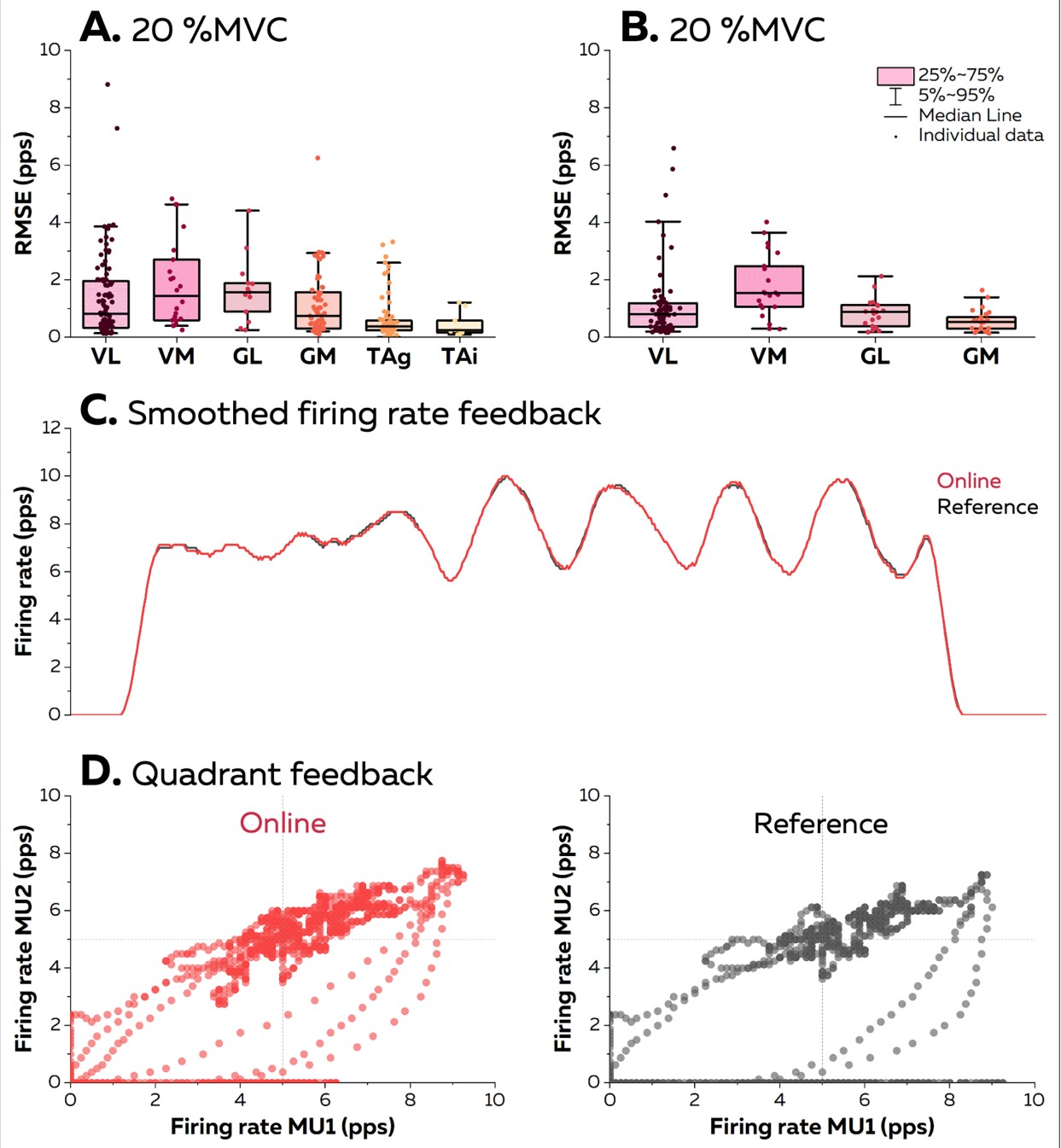

**Figure 6.** Accuracy of the visual feedback based on the motor unit smoothed firing rates. The online electromyographic (EMG) decomposition was provided to the participants in the form of different visual feedback. We estimated the accuracy of the feedback by calculating the root mean squared error (RMSE) between the smoothed firing rates estimated from the motor unit spike trains identified in real time and from their manually edited version at 20% maximal voluntary contraction (MVC) (**A**) and 40% MVC (**B**). Each dot is an individual motor unit, each box represents the 25th and 75th percentiles of the distribution of values, each bar represents the 5th and 95th percentiles of the distribution of values, and each line is the median. VL: vastus lateralis, VM: vastus medialis, GL: gastrocnemius lateralis, GM: gastrocnemius medialis, TAg: tibialis anterior recorded with a high-density grid of surface electrodes, and TAi: tibialis anterior recorded with an intramuscular array of electrodes. (**C**) Smoothed firing rate of a motor unit provided in real time (Real time, red) vs. its manually edited version (Reference, grey). (**D**) Quadrant plot provided in real time (Real time, red) vs. its manually edited version (Reference, grey). Each dot represents the position of the cursor during one window of 125 ms.

contractions with a stable level of force, as even large changes in force during isometric contractions can impact the orientation of the muscle fibres relatively to the skin within muscles (*Ito et al., 1998*).

## Usage of the software

The first step of the algorithm consists of identifying motor unit separation vectors (motor unit filters) from EMG signals with fastICA (*Negro et al., 2016*; *Barsakcioglu et al., 2021*; *Chen and Zhou, 2016*). Classically, metrics such as the Pulse to Noise ratio (*Holobar et al., 2014*) or the silhouette value (*Negro et al., 2016*) are used to assess the reliability of the identified motor units by estimating the distance between the spikes and the noise. Here, we purposely reported all the motor units identified from offline decomposition in *Figure 2* to illustrate the importance of manual editing, but a threshold should be used to automatically remove inaccurate spike trains (i.e. 0.9 in *Negro et al., 2016*). It is noteworthy that such approach must always be associated with extensive manual editing to remove the discharge times incorrectly selected and to add missing spikes (*Del Vecchio et al., 2020*; *Hug et al., 2021*). This necessary step precedes the update of the motor unit filter with all the detected spikes (*Holobar et al., 2010*), leading to an increase in the silhouette value and a decrease in the coefficient of variation of the interspike intervals (*Figure 2B*). Alternatively, one could speed up the processing flow by setting a higher threshold for the silhouette value. This would decrease the burden of manual editing at the cost of decreasing the number of identified motor units. However, it is important to note that using more stringent criteria does not preclude the manual editing, even for motor units with a high silhouette value (see motor units 1 and 2 with missed spikes in *Figure 1B*).

The second step of the algorithm consists of identifying motor units discharge times in real time by projecting extended segments of EMG signals on separation vectors (motor unit filters) to estimate motor unit pulse trains (*Figure 1B*; *Barsakcioglu et al., 2021*; *Chen et al., 2020*). This method was effective at identifying motor unit discharge times, with rates of agreement >0.75, regardless of the contraction intensity and muscle (*Figure 5*). It is noteworthy that the performance was particularly high for the recordings made with an intramuscular array of electrodes (rate of agreement of 0.97 ± 0.05, *Figure 5*). As mentioned above, the better performance of blind-source separation on multichannel intramuscular EMG has already been reported with offline analyses (*Muceli et al., 2015*; *Muceli et al., 2022*; *Chen et al., 2019*). This is explained by the higher spatial selectivity of the electrodes (*LeFever and De Luca, 1982*), the larger bandwidth of the signal (*Lindstrom and Magnusson, 1977*), and the higher robustness of the motor unit filter as the signal is less affected by the geometric changes of the volume conductor. In contrast, precision and rates of agreements were lower for motor units identified over the vastii and gastrocnemii muscles when compared to the TA muscle (*Figure 5*). Even though the reason for this between-muscle difference is unclear, it is possible that the specific activation level of the vastii and gastrocnemii muscles varied more than that of TA between the baseline and test contractions because of muscle redundancy leading to multiple coordination strategies possible to perform knee flexion or plantarflexion. For example, a decrease in activation would decrease the height of the peaks of the estimated sources, potentially classified in the noise class during the online decomposition (*Chen et al., 2020*). Conversely, an increase in activation may activate motor units spatially close to those observed during the baseline contraction, corrupting their pulse train with merged sources (*Holobar and Farina, 2014*; *Farina and Holobar, 2016*). This explanation is in line with the lower rate of agreement observed when participants tracked a force target higher than the level of the baseline contraction (*Figures 3 and 4*). One way to overcome these challenges would be to dynamically update the motor unit filters and the centroids of the 'spikes' and 'noise' classes (*Chen et al., 2020*; *Yeung et al., 2024*; *Mendez Guerra et al., 2024*). While appealing, this approach is also computationally demanding (*Chen et al., 2020*; *Yeung et al., 2024*; *Mendez Guerra et al., 2024*). We propose to update the motor unit filters and the centroids of the 'spikes' and 'noise' classes during the resting periods. In addition, it is recommended that the operator displays the motor unit pulse trains and identified discharge times between contractions to check for decomposition accuracy across the session.

## Inter-individual differences in motor unit yields

An important consideration regarding the implementation of offline or real-time surface EMG decomposition is the difference between individuals, with an overall lower yield in number of identified motor units in females (here: 9 ± 12) than in males (here: 30 ± 13). Typically, the number of identified

motor units from surface EMG is twice as low in females than males (*Del Vecchio et al., 2020*; *Lulic-Kuryllo and Inglis, 2022*; *Taylor et al., 2022*). The cause for this difference remains unclear. It may be related to variations in properties of the tissues separating the motor units from the recording electrodes, or to differences in the morphological and physiological properties of muscle fibres, as well as to the innervation ratios of motor units. These sex-related differences have so far only been supported by data extracted from animal experiments (*Mierzejewska-Krzyżowska et al., 2011*). However, the recent developments of simulation frameworks capable of generating highly realistic EMG signals for anthropometrically diverse populations may help understanding the impact of sex-related differences in humans (*Maksymenko et al., 2023*). Specifically, these simulations can account for diverse anatomical (e.g. muscle volume and architecture, thickness of subcutaneous tissues) and physiological characteristics (e.g. innervation ratio, number of motor units, fibre cross-sectional area, fibre conduction velocity, contribution of rate coding vs. spatial recruitment). Generating such dataset could help identifying the primary factors affecting EMG decomposition performance, ultimately enabling the refinement of algorithms and/or surface electrode design.

## Summary

Overall, the main purpose of our software is to display to the participant a real-time visual feedback based on the activity of individual motor units or populations of motor units. The RMSE of the smoothed discharge rates was constantly below two pulses per second, with values as low as 0.44 ± 0.43 pps for the TA muscle recorded with intramuscular electrode arrays. Thus, the movement of the cursors accurately represented the variations in motor unit firing activity to the participant. In this study we have presented results of control of smoothed firing rates over a relatively large smoothing window, but the duration of the smoothing filter can be chosen by the user according to the needs and applications. Operators could use the provided software to interact with a virtual environment, such as typing on a keyboard with a cursor moved by modulating motor unit firing rates (*Formento et al., 2021*). In the field of motor control, neuroscientists may train participants to selectively activate individual motor units (*Bräcklein et al., 2022*), testing the concept of rigid versus flexible motor control (*Formento et al., 2021*; *Bräcklein et al., 2022*; *Marshall et al., 2022*), or movement augmentation (*Eden et al., 2022*). Generally, we hope that this open-source software will open perspectives for neuroscientists to design experimental paradigms that takes advantage of online EMG decomposition to study the neural control of movements at the motor neuron level.

## Materials and methods
### Simulation of EMG signals

We generated synthetic EMG signals using an anatomical model of a cylindrical muscle volume with parallel fibres surrounded by subcutaneous tissues and skin (*Farina et al., 2008a*; *Konstantin et al., 2020*). The muscle radius was 10 mm, and the thicknesses of the subcutaneous and skin layers were 4 mm and 1 mm, respectively. The centres of 150 motor unit territories were randomly and evenly distributed across the muscle cross-section. The number of fibres innervated by each motor neuron followed an exponential distribution, ranging from 66 to 3321 (*Enoka and Fuglevand, 2001*). The fibres of the same motor unit were positioned around the centre of the motor unit within areas of 0.8–78.5 mm². The fibre density in the muscle reached 400 fibres/mm². The motor unit action potentials were detected in the model by either a grid of 64 circular surface electrodes with a diameter of 1 mm arranged in 5 columns and 13 rows (inter-electrode distance: 4 mm) or 16 circular intramuscular electrodes arranged in a single array (inter-electrode distance: 1 mm). The grid was centred over the muscle in the transverse direction. The intramuscular electrode array was centred with respect to the grid and inserted at an angle of 30° along the longitudinal direction of the volume toward the centre of the muscle.

Four profiles of force were generated, with either trapezoidal patterns reaching 10, 20, or 30% of the maximal force, or a sinusoidal pattern with the force varying between 15 and 25% MVC at a rate of 0.5 Hz. Each discharge time elicited single-fibre electrical potentials, summed together within each motor unit. The profile of each motor unit action potential changed between electrodes according to the morphology of the fibres and the distance of the fibres relative to the electrode. Ultimately, the synthetic EMG signal represented the sum of all motor unit action potentials and Gaussian noise.

## Electromyographic recordings

The accuracy and capabilities of the online EMG decomposition algorithm was tested on a series of experimental data collected with either surface or intramuscular arrays of electrodes from five different muscles.

Surface EMG signals were recorded from the gastrocnemius medialis and lateralis (GM and GL; triceps surae protocol) or the vastus medialis and lateralis (VM and VL; quadriceps protocol) with two-dimensional adhesive grids of 64 electrodes (13 × 5 electrodes with one electrode absent on a corner, gold-coated, interelectrode distance: 8 mm; GR08MM1305, OT Bioelettronica). Surface EMG signals were recorded from the tibialis anterior with four two-dimensional adhesive grids of 64 electrodes (13 × 5 electrodes with one electrode absent on a corner, gold-coated, interelectrode distance: 4 mm; GR04MM1305, OT Bioelettronica). Before the placement of the grids, the skin was shaved and cleaned with an abrasive gel (Nuprep, Weaver and Company, USA). Each adhesive grid was held on the skin using a disposable adhesive foam layer. The cavities within the adhesive layer were filled with conductive paste (SpesMedica, Italy) to facilitate the skin-electrode contact. A 10-cm-wide elastic band was placed over the electrodes to ensure good contact between the electrodes and the skin throughout the experiment.

An intramuscular linear array of 16 electrodes on a thin-film (platinum-coated, interelectrode distance: 1 mm) was inserted into the tibialis anterior in one participant at an approximate angle of 30°. The insertion was guided with a portable ultrasound probe (Butterfly IQ+, Butterfly Network, USA).

A reference electrode was positioned over the tibia of the right limb (triceps surae protocol), over the patella of the right limb (quadriceps protocol), or over the medial malleolus (tibialis anterior protocol). A strap electrode dampened with water was placed around the ankle (ground electrode) for each data collection. The EMG signals were recorded in monopolar mode and digitized together with the torque signal at a sampling rate of 2048 Hz for the grids of surface electrodes and 10,240 Hz for the intramuscular array of electrodes (EMG-Quattrocento, 400 channel EMG amplifier; OT Bioelettronica).

## Experimental procedure

A total of 29 individuals participated in the experiments (4 females/25 males; 28 ± 5 years old; 178 ± 6 cm; 73 ± 16 kg). None of the participants reported lower limb injury or pain in the 6 months prior to testing. Ethical committees approved the study (triceps surae and quadriceps protocols: CERNI – Nantes Université, n°04022022; tibialis anterior protocol: Imperial College London, no. 18IC4685). All participants provided their informed written consent before the beginning of the experiment.

The right side of the body was tested for all participants and for all protocols. For the triceps surae protocol, participants sat on a dynamometer (Biodex System 3 Pro, Biodex Medical, USA) with their hip flexed at 80°, 0° being the neutral position, and their right leg fully extended. Their ankle angle was set to 10° of plantarflexion, 0° being the foot perpendicular to the shank. For the quadriceps protocol, participants sat on the dynamometer with their hips flexed at 80° and the knee of their right leg flexed at 80°, 0° being the full extension. Inextensible straps were tightened during both tasks to immobilise the torso, pelvis, and thigh on the test side. For the tibialis anterior protocol, participants sat on a chair while their foot was fixed onto the pedal of a dynamometer (OT Bioelettronica) coupled with a load cell (CCT Transducer, Italy) and positioned at 30° in the plantarflexion direction (0° being the foot perpendicular to the shank). The foot was fixed to the pedal with inextensible straps positioned around the proximal phalanx, metatarsal, and cuneiform. Force signals were recorded using the same acquisition system as for the EMG recordings (EMG-Quattrocento; OT Bioelettronica).

All experiments began with a standardised warm up, which included five 3 s isometric plantar flexion or knee extension contractions at 50, 60, 70, and 80%, and three 3 s contractions at 90% of the participants' subjective maximal torque. Then, after 2 min of rest, participants performed three MVCs for 3–4 s, with 60 s of rest in between. Peak MVC torque was considered as the maximal value obtained from a moving average window of 250 ms.

For the triceps surae and quadriceps protocols, participants performed three trapezoid isometric contractions at 40% of the MVC (referred to as baseline contractions) to identify motor unit filters offline. Each of these contractions involved a 5 s ramp-up, a 20 s plateau, and a 5 s ramp-down phase and was separated by 60 s of rest. The separation vectors were identified offline from these

contractions and then applied in real time to estimate the firing activity of each identified motor unit during three additional trapezoid contractions (referred to as the online task). Each of these online tasks involved a 5 s ramp-up, a 30 s plateau, and a 5 s ramp-down phase and was separated by 60 s of rest. To test the effect of variations in contraction intensity between the online task and the baseline contraction used to identify the separation vectors, each plateau consisted of three successive 10 s targets at 35, 40, or 45% of the MVC performed in a random order. During these online tasks, feedback of motor unit discharge times and torque output was displayed in real time on a monitor to the participants.

To test the effect of the baseline contraction intensity on the accuracy of real-time identification of motor unit discharge activity; the procedure was repeated with a baseline intensity of 20% MVC. During the last three trapezoidal contractions, each plateau consisted of three successive 10 s targets at 15, 20, or 25% of the MVC performed in a random order.

For the tibialis anterior protocol with grids of surface electrodes, participants performed a trapezoid contraction at 10 and/or 20% of the MVC, involving a 10 s ramp up, a 60 s plateau, and a 10 s ramp-down phase (baseline contraction). The separation vectors were identified from this contraction and then applied in real time over a second identical contraction (online task). The same procedure was repeated for the tibialis anterior protocol with an intramuscular array of electrodes, with contractions involving a ramp up phase of 2 s, a plateau of 20 s, and a ramp-down phase of 2 s.

## Data processing
### Mathematical modelling of the recorded spike trains

The spike train of a motor neuron recorded over time $t \in [0, T]$ can be described as the result of a convolution between a delta function ($\delta$) representing the firing times ($\varphi$), and finite impulse responses ($h$) representing action potentials of duration $L : \sum_{l=0}^{L-1} h(l) \sum_r \delta(t - \phi_r - l)$. In practice, the nature of $h$ and the duration L depend on the type of recordings. For electrophysiological measurements, $h$ characterises the local electrical field generated by the spike and conducted through the surrounding tissues.

As the recorded volume of tissue comprises many active neurons, each recording can be considered as a convolutive mixture of multiple sources, and the previous equation can be expressed in the form of a matrix to also consider all the electrodes of an array: given $s(k) = \sum_r \delta(k - \varphi_r); X(t) = \sum_{l=0}^{L-1} H(l)S(k - l) + N(t)$, where $X(t) = [x_1(t), x_2(t), ..., x_m(t)]^T$ is a matrix of $m$ electrophysiological signals, $S(t) = [s_1(t), s_2(t), ..., s_n(t)]^T$ is a matrix of $n$ motor neurons' spike trains, and $H(l)$ is a $m$ by $n$ matrix containing the $l$th sample of action potentials from $n$ neurons and $m$ signals. In this situation, we can reformulate the model as an instantaneous mixture of an extended set of sources, that is, the motor neurons' spike trains and their delayed versions. This allows us to simply write the previous equation as a multiplication of matrices, in which each source is delayed L times, L being the duration of the impulse response $h$. This model can be inverted for neural decoding with source-separation approaches.

### Identification of separation vectors (motor unit filters) with blind-source separation

The monopolar EMG signals collected during the baseline contractions were extended with an extension factor of $\frac{1000}{m}$ (**Holobar and Zazula, 2007**), where $m$ is the number of channels free of any noise or artefact. The signals were then demeaned and whitened. A contrast function was iteratively applied to estimate a separation vector that maximised the level of sparseness of the motor unit pulse train (**Figure 1B**). This loop stopped when the variation of the separation vector between two successive iterations reaches a predefined lower bound. After the application of a peak detection algorithm, the motor unit pulse train contained high peaks (i.e. the spikes from the identified motor unit) and low peaks from other motor units and noise. High peaks were separated from low peaks and noise using k-mean classification with two classes (**Figure 1B**). The peaks from the class with the highest centroid were considered as spikes of the identified motor unit. A second algorithm refined the estimation of the discharge times by iteratively recalculating the separation vector and repeating the steps with peak detection and k-mean classification until the coefficient of variation of the interspike intervals was minimised. The accuracy of each estimated spike train was assessed by computing the silhouette (SIL) value between the two classes of peaks identified with k-mean classification (**Negro et al., 2016**).

When the SIL exceeded a predetermined threshold, the motor unit filter was saved for the real-time decomposition, together with the centroids of the 'spikes' and 'noise' classes (*Figure 1B*).

## Manual editing

There is a consensus among experts that automatic decomposition should be followed by visual inspection and manual editing (*Martinez-Valdes et al., 2023*). Manual editing involves the following steps: (1) removing spikes that result in erroneous firing rates (outliers), (2) adding discharge times that are clearly distinguishable from the noise, (3) recalculating the separation vector, (4) reapplying the separation vector on the EMG signals (either a selected window or the entire signal), and (5) repeating this procedure until no outliers are present and all clearly distinguishable spikes have been selected. Importantly, the manual editing of potentially missed or falsely identified discharge times should not be accepted before the application of the updated motor unit separation vector, thereby generating a new pulse train. Manual edits should be accepted only if the silhouette value improves following this operation or remains well above the pre-established threshold. A more extensive description of the manual editing of motor unit pulse trains can be found in *Del Vecchio et al., 2020*. Even though some of the aforementioned steps involve subjective decision-making, evidence suggests that manual editing after EMG decomposition with blind-source separation approaches remains highly reliable across operators (*Hug et al., 2021*). Specifically, the median rate of agreement calculated for 126 motor units over eight operators with various experience in manual editing was 99.6%. All raw and processed data have been made available on a public data repository so that they can be used for training new operators (http://doi.org/10.6084/m9.figshare.13695937).

## Real-time identification of motor neuron activities

EMG signals were transmitted by packages of 256 data points for the surface grid recordings (125 ms with a sampling frequency of 2048 Hz) or 1280 data points for the intramuscular array recordings (125 ms with a sampling frequency of 10,240 Hz). The *electrode mask* determined from the baseline contractions was applied to remove the channels with noise or artefacts and the data was extended using the same extension factor as for the baseline contraction (see the section on EMG decomposition above). The matrix of separation vectors identified during the baseline contraction was applied over the extended EMG signal. Local peaks were identified for each motor unit using the MATLAB function 'islocalmax' with a minimal separation of 25 ms between peaks to limit the number of false positives (*Figure 1C*). These peaks were considered as spikes when their distance from the centroid of the 'spike' class was shorter than the distance from the centroid of the 'noise' class (*Figure 1C*). Both centroids were identified from the offline decomposition made on the baseline contraction.

We calculated the firing rate (i.e. spikes per second) of individual motor units as the sum of the spikes over a moving window of eight consecutive epochs of 125 ms. We chose this approach to facilitate the smoothness of the visual feedback, in contrast to the instantaneous firing rate, which is calculated as the instantaneous inversed interspike interval, which oscillates more due to the presence of synaptic noise. While this approach introduces a delay of 500 ms in the estimation of the firing rate, we identified in pilot testing that this also facilitates the control of motor unit firing activities by the participant. To further increase the smoothness of the online biofeedback, we added the option in the software to average individual firing rates using a moving window. For the data reported in this article, we selected a value of four consecutive values, corresponding to a window of 500 ms. Note that researchers who aim to minimise the delay of the visual feedback can disable this option, change its value, or use the real-time raster plots that displays the instantaneous discharge times of each motor unit.

## Computational time

The computational time depends on the number of identified motor units during the baseline contraction, the number of peaks sorted during each epoch, and the number of EMG channels retained for the analysis. We considered the computational time for the decomposition as the time between the reception of the EMG signals by the computer and the estimation of the discharge times of all the identified motor units. We considered the computational time for the feedback as the time between the decomposition of the EMG signals and the update of the online feedback. The computational times were calculated on a laptop equipped with an Apple M1 Max chip and 64 GB of RAM.

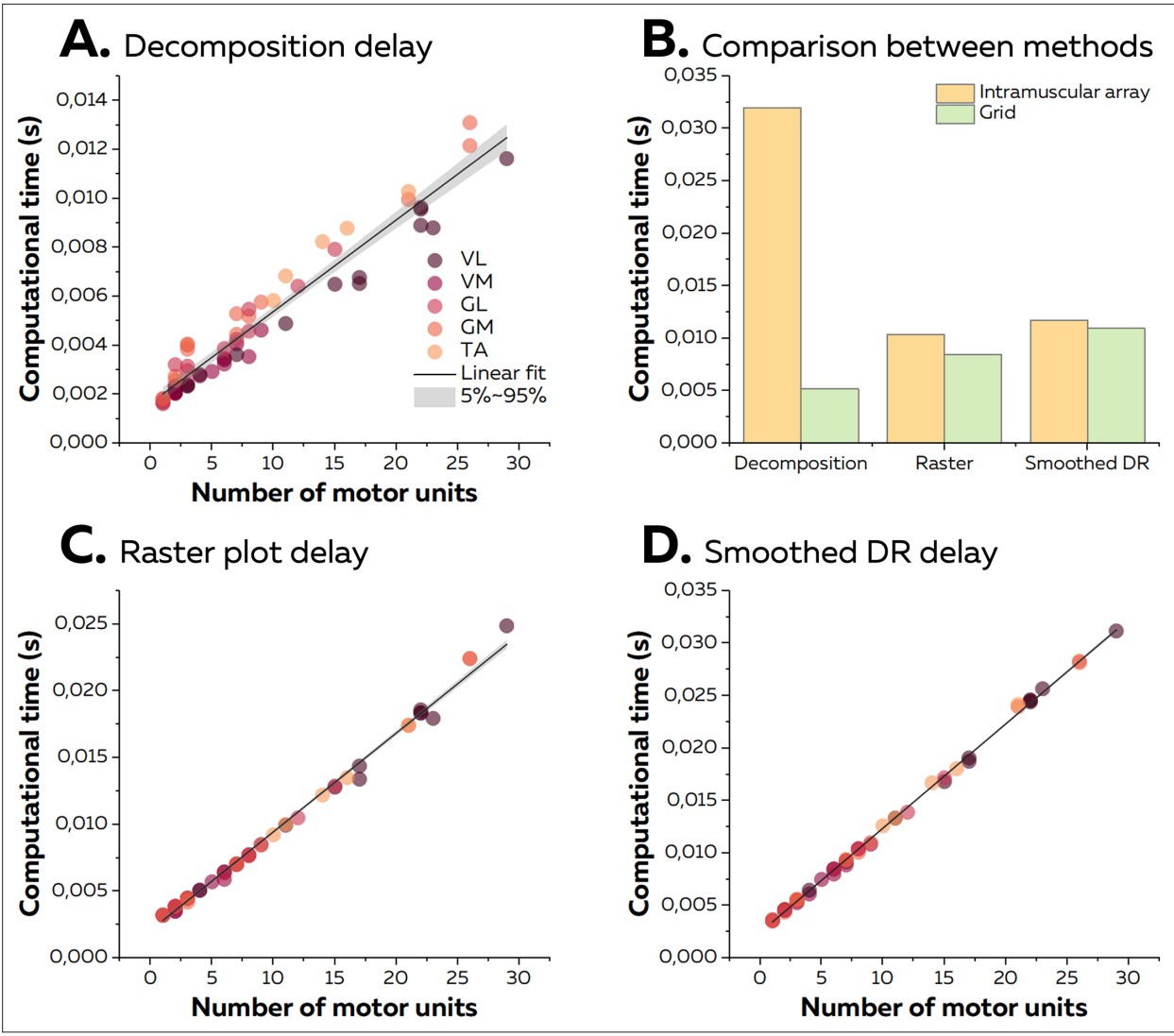

**Figure 7.** Computational time of the online electromyographic (EMG) decomposition. (**A**) We considered the computational time for the decomposition as the time between the reception of the EMG signals and the identification of the spikes for all the motor units. We computed the linear regression between the number of identified motor units and the computational time and considered the slope as the computational time per motor unit. Each dot represents one decomposition, and the colour scheme depends on the muscle. (**B**) As the sampling frequency differed between recordings with high-density grids and intramuscular arrays of electrodes, we compared the computational times for both techniques with the same number of identified motor units (i.e. 9). (**C, D**) After the decomposition, the motor unit discharge activity was translated into visual feedback, either in the form of a raster plot or the smoothed firing rates for all the identified motor units. As for the decomposition, we normalised the computational time per motor unit using a linear regression.

On average, the time between the reception of an epoch of EMG signals and the identification of the spikes was 4.9 ± 3.0 ms for data collected with high-density grids of surface electrodes. To estimate the computational time per motor unit, we performed a linear regression between the number of identified motor units and the computational time per epoch using data from all the experiments made with high-density surface electrodes. The slope of the linear fit, that is, the computational time per motor unit, was 0.37 ms (*Figure 7A*). Across all the experiments, the maximal computational time was 13.1 ms for 26 identified motor units. Of note, when considering the experiment made with intramuscular arrays of electrodes, the computational time reached 32.0 ms for nine identified motor units. This was longer than observed for the same number of identified motor units using high-density surface electrodes (5.2 ms; *Figure 7B*) and was due to the greater sampling frequency used for the intramuscular recordings.

We then calculated the additional computational time to display visual feedback in the form of a raster plot or smoothed firing rates of all the identified motor units. On average, we observed computational times of 8.6 ± 5.7 ms and 11.2 ± 7.6 ms for raster plots and smoothed firing rates, respectively. The computational time per motor unit was 0.74 ms and 0.99 ms for raster plots and smoothed firing rates, respectively (*Figure 7*). Of note, the computational time for the quadrant plot made from the activity of two motor units reached 14.8 ± 0.0 ms on average. The standard deviation of computational times across windows reached 5.4 ± 4.0 ms (raster plot), 4.0 ± 3.2 ms (smoothed firing rate), and 2.8 ± 2.5 ms (quadrant). It is noteworthy that the computational times for experiments with grids of electrodes or intramuscular arrays of electrodes was similar regardless of the type of visual feedback (*Figure 7*). Overall, as the total computational time was constantly shorter than the duration of an epoch of EMG signals, the visual feedback was always updated during the recording of the next epoch of EMG signals. Therefore, the only delay was the incompressible recording time per epoch of signals, that is, 125 ms.

## Accuracy of the real-time identification of motor unit firing activity

To assess the accuracy of the real-time identification of motor unit spike trains, we compared the motor unit spike trains identified in real time with those obtained after manual editing. The manual editing was performed offline as described above (*Del Vecchio et al., 2020*; *Hug et al., 2021*).

The accuracy of the real-time decomposition was assessed for each motor unit by computing the sensitivity [TP/(TP + FN)], the precision [TP/(TP + FP)], the false negative rate [FN/(TP + FN)], and the rate of agreement [TP/(TP + FN + FP)] between the manually edited spike train (offline) and the spike train identified in real time.

Here, TP (true positive) is the number of spikes identified in both the real time and edited spike trains, FP (false positive) is the number of spikes only identified in the real-time spike train and FN (false negative) is the number of spikes only identified in the edited spike train.

To assess the accuracy of the biofeedback provided by the software, we measured the RMSE between the path drawn by the smoothed firing rate of motor units estimated in real time and the path estimated from the manually edited motor unit spike trains.

## Data availability

The entire data set (raw and processed data) codes and a user manual of the software are available at https://github.com/simonavrillon/I-Spin, copy archived at *Avrillon, 2024*.

# Additional information

### Competing interests

Dario Farina: inventor of a patent (Neural Interface. UK Patent application no. GB1813762.0. August 23, 2018) and of a patent application (Neural interface. UK Patent application no. GB2014671.8. September 17, 2020), which are partly related to the methods and applications of this work. All other authors have no financial or other relationships that might lead to a conflict of interest. The other authors declare that no competing interests exist.

### Funding

| Funder | Grant reference number | Author |
| --- | --- | --- |
| Agence Nationale de la Recherche | ANR-15-IDEX-01 | François Hug |
| European Research Council | #810346 | Dario Farina |
| Engineering and Physical Sciences Research Council | EP/T020970 | Dario Farina |
| Biotechnology and Biological Sciences Research Council | BB/V00896X | Dario Farina |

| Funder | Grant reference number | Author |
|---|---|---|

The funders had no role in study design, data collection and interpretation, or the decision to submit the work for publication.

## Author contributions

Julien Rossato, Conceptualization, Data curation, Software, Formal analysis, Validation, Investigation, Methodology, Writing – original draft, Writing – review and editing; François Hug, Conceptualization, Resources, Supervision, Funding acquisition, Validation, Investigation, Methodology, Writing – original draft, Writing – review and editing; Kylie Tucker, Conceptualization, Supervision, Writing – original draft, Writing – review and editing; Ciara Gibbs, Software, Validation, Methodology, Writing – review and editing; Lilian Lacourpaille, Supervision, Writing – review and editing; Dario Farina, Resources, Writing – review and editing; Simon Avrillon, Conceptualization, Data curation, Software, Formal analysis, Supervision, Validation, Investigation, Methodology, Writing – original draft, Writing – review and editing

## Author ORCIDs

Dario Farina (i) https://orcid.org/0000-0002-7883-2697
Simon Avrillon (i) https://orcid.org/0000-0002-2226-3528

## Ethics

Ethical committees approved the study (triceps surae and quadriceps protocols: CERNI - Nantes Université, no. 04022022; tibialis anterior protocol: Imperial College London, no. 18IC4685). All participants provided their informed written consent before the beginning of the experiment.

Reviewer #1 (Public review): https://doi.org/10.7554/eLife.88670.3.sa1
Reviewer #3 (Public review): https://doi.org/10.7554/eLife.88670.3.sa2
Author response https://doi.org/10.7554/eLife.88670.3.sa3

# Additional files

## Supplementary files

• MDAR checklist

## Data availability

All data recorded for this study are available on Figshare. Code and apps associated with this study are available on Github at https://github.com/simonavrillon/I-Spin (copy archived at *Avrillon, 2024*).

The following datasets were generated:

| Author(s) | Year | Dataset title | Dataset URL | Database and Identifier |
|---|---|---|---|---|
| Avrillon S | 2024 | TA - thin films | https://doi.org/10.6084/m9.figshare.26984026 | figshare, 10.6084/m9.figshare.26984026 |
| Avrillon S | 2024 | TA - grids | https://doi.org/10.6084/m9.figshare.26983384 | figshare, 10.6084/m9.figshare.26983384 |
| Avrillon S | 2024 | GL GM - grids | https://doi.org/10.6084/m9.figshare.26983615 | figshare, 10.6084/m9.figshare.26983615 |
| Avrillon S | 2024 | VL VM - grids | https://doi.org/10.6084/m9.figshare.26983963 | figshare, 10.6084/m9.figshare.26983963 |

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
