## [Editor Report · eLife assessment]

This article compiles existing algorithms into an open-source software package that enables real-time (and offline) motor unit decomposition from muscle activity collected via grids of surface electrodes and indwelling electrode arrays. The package is **valuable** given that many motor neuroscience labs are using such algorithms and that there exists a host of potential applications for such data. Validation of the software package is **compelling**, suggesting that it can be successfully applied across a range of muscles and tasks.

---

## [Referee Report · Reviewer #1 (Public review)]

Many labs world-wide now use the blind source deconvolution technique to identify the firing patterns of multiple motor units simultaneously in human subjects. This technique has had a truly transformative effective on our understanding of the structure of motor output in both normal subjects and, increasingly, in persons with neurological disorders. The key advance presented here is that the software provides real time identification of these firing patterns.

The main strengths are the clarity of the presentation and the great potential that real-time decoding will provide. Figures are especially effective and statistical analyses are excellent.

---

## [Referee Report · Reviewer #3 (Public review)]

In this manuscript, Rossato and colleagues present a method for real-time decoding of EMG into putative single motor units. Their manuscript details a variety of decision points in their code and data collection pipeline that lead to a final result of recording on the order of ~10 putative motor units per muscle in human males. Overall the manuscript is highly restricted in its potential utility but may be of interest to aficionados. For those outside the field of human or nonhuman primate EMG, these methods will be of limited interest.

Comment on revised version

The revised manuscript has thoroughly and responsively addressed the concerns and suggestions raised in the first review. I think the method will be of use to the field and fits well within the purview of eLife's publications on methods development.

---

## [Author Response]

The following is the authors’ response to the original reviews.

**eLife assessment**
This manuscript compiles existing algorithms into an open-source software package that enables realtime motor unit decomposition from muscle activity collected via grids of surface electrodes and indwelling electrode arrays. The software package is valuable given that many motor neuroscience labs are using such algorithms and that there exist a host of potential real-time applications for such data. Validation of the software package is generally solid but incomplete in some important areas: the primary data is narrow in scope and only from male participants, and there is a lack of ground truth tests on synthetic data. The impact of the software package could be strengthened by making it less tied to specific electrode hardware and by expanding it to easily permit offline analysis.

We thank the reviewers and editors for their comments and suggestions after reading the initial version of our manuscript. In this second iteration, we have performed a validation of the algorithm using synthetic EMG signals. We have also added experimental data collected in female participants. Finally, the new version of I-Spin is compatible with the Open Ephys GUI that can interface with devices such as the Open Ephys and Intan acquisition boards. Another version has been developed for interfacing with the devices provided by the TMSi company (https://info.tmsi.com/blog/ispin-saga-real-timemotor-unit-decomposition-tool). We believe that such changes will make I-Spin more accessible for a broad range of experimental setups and research teams. Please find below the specific answers to the reviewers’ comments.

**Reviewer #1 (Public Review):**
Many labs worldwide now use the blind source deconvolution technique to identify the firing patterns of multiple motor units simultaneously in human subjects. This technique has had a truly transformative effect on our understanding of the structure of motor output in both normal subjects and, increasingly, in persons with neurological disorders. The key advance presented here is that the software provides real-time identification of these firing patterns. The main strengths are the clarity of the presentation and the great potential that real-time decoding will provide. Figures are especially effective and statistical analyses are excellent.

We thank the reviewer for this positive appreciation of our work.

The main limitation of the work is that only male subjects were included in the validation of the software. The reason given - that yield of number of motor units identified is generally larger in males than females - is reasonable in the sense that this is the first systematic test of this real-time approach. At a minimum, however, the authors should clearly commit to future work with female subjects and emphasize the importance of considering sex differences.

As emphasised by the reviewer, the number of identified motor units is typically higher in males than females when using surface EMG (Taylor et al., 2022), which is the current main limitation of the implementation of offline EMG decomposition technique in a broad and representative sample of research participants. These differences between biological sex are less present when using intramuscular EMG, as the signals are less affected by the filtering effect of the volume conductor separating the motor units from the recording electrodes. Besides the different yields expected between males and females, we do not expect differences in terms of the accuracy of the motor unit identification algorithm, which is the main outcome of this paper.

Nevertheless, we acknowledge the importance to understand the reasons for this difference, and the imperative to refine algorithms and/or surface electrode design to mitigate this major limitation with surface EMG.

To support this point, the discussion has been updated (P20; L480):

‘An important consideration regarding the implementation of offline or real-time surface EMG decomposition is the difference between individuals, with an overall lower yield in number of identified motor units in females (here: 9 ± 12) than in males (here: 30 ± 13). Typically, the number of identified motor units from surface EMG is twice as low in females than males (32, 49, 50). The cause for this difference remains unclear. It may be related to variations in properties of the tissues separating the motor units from the recording electrodes, or to differences in the morphological and physiological properties of muscle fibres, as well as to the innervation ratios of motor units. These sex-related differences have so far only been supported by data extracted from animal experiments (51). However, the recent developments of simulation frameworks capable of generating highly realistic EMG signals for anthropometrically diverse populations may help understanding the impact of sex-related differences in humans (52). Specifically, these simulations can account for diverse anatomical (e.g. muscle volume and architecture, thickness of subcutaneous tissues) and physiological characteristics (e.g. innervation ratio, number of motor units, fibre cross sectional area, fibre conduction velocity, contribution of rate coding vs. spatial recruitment). Generating such dataset could help identifying the primary factors affecting EMG decomposition performance, ultimately enabling the refinement of algorithms and/or surface electrode design.’

Finally, we have completed new experiments including males and females in this new iteration (P.12; L.295):

‘Application of motor unit filters in experimental data

We then asked eight participants (4 males and 4 females) to perform trapezoidal isometric contractions with plateaus of force set at 10% and 20% MVC during which surface EMG signals were recorded from the TA with 256 electrodes separated by 4 mm. The aim of this experiment was to confirm the results of the simulation; specifically, to test the accuracy of the online decomposition when the level of force was below, equal to, or above the level of force produced during the baseline contraction used to estimate the motor unit filters (Figure 4). We assessed the accuracy of the motor unit spike trains identified in real time using their manually edited version as reference. 144 motor units were identified at both 10 and 20% MVC. When the test signals were recorded at the same level of force as the baseline contraction, we obtained rates of agreement of 95.6 ± 6.8% (10% MVC) and 93.9 ± 5.9% (20% MVC). The sensitivity reached 95.9 ± 6.7% (10% MVC) and 94.4 ± 5.6% (20% MVC), and the precision reached 99.6 ± 1.3% (10% MVC) and 99.4 ± 1.9% (20% MVC).

When the filters identified at 20% MVC were applied on signals recorded at a lower level of force (10% MVC), the rates of agreement decreased to 87.9 ± 16.2%. The sensitivity also decreased to 88.0 ± 16.2%, but the precision remained high (99.4 ± 4.3). Thus, the decrease in accuracy was mostly caused by missed discharge times rather than the false identification of artifacts or spikes from other motor units. When the filters identified at 10% MVC were applied to signals recorded at a higher level of force, the rates of agreement decreased to 83.3 ± 13.5%. The sensitivity decreased to 90.7 ± 8.1%, and the precision also decreased to 90.9 ± 12.6%. This result confirms what was observed with synthetic EMG, that is motor units recruited between 10 and 20% MVC can substantially disrupt the accuracy of the decomposition in real-time, as highlighted in Figure 4 (lower panel). Importantly, this situation does not happen for all the motor units, as suggested by the distribution of the values in Figure 4.’

A second weakness is that the Introduction does a poor job of establishing the potential importance of the real-time approach.

The introduction has been modified to highlight the importance of identifying the spiking activity of motor units in real time. Specifically, the first paragraph has been rewritten to read (P3; L67):

‘The activity of motor neuron – in the form of spike trains – represents the neural code of movement to muscles. Decoding this firing activity in real-time during various behaviours can thus substantially enhance our understanding of movement control (2-5). Real-time decoding is also essential for interfacing with external devices (6) or virtual limbs (7) when activity is present at the periphery of the nervous system. For example, individuals with a spinal cord injury can control a virtual hand with the residual firing activity of the motor units in their forearm (7). Furthermore, sampling the activity of motor units receiving a substantial portion of independent synaptic inputs may pave the way for movement augmentation – specifically, extending a person’s movement repertoire through the increase of controllable degrees of freedom (8). In this way, Formento et al. (3) showed that individuals can intuitively learn to independently control motor units within the same muscle using visual cues. Having access to open-source tools that perform the real-time decoding of motor units would allow an increasing number of researchers to improve and expand the range of these applications’

**Reviewer #2 (Public Review):**
Rossato et al present I-spin live, a software package to perform real-time blind-source separation-based sorting of motor unit activity. The core contribution of this manuscript is the development and validation of a software package to perform motor unit sorting, apply the resulting motor unit filters in real-time during muscle contractions, and provide real-time visual feedback of the motor unit activity. I have a few concerns with the work as presented:I found it challenging to specifically understand the technical contributions of this manuscript. The authors do not appear to be claiming anything novel algorithmically (with respect to spike sorting) or methodologically (with respect to manual editing of spikes before the use of the algorithms in real-time). My takeaway is that the key contributions are C1 development of an open-source implementation of the Negro algorithm, C2 validating it for real-time application (evaluating its sorting efficacy, and closed-loop performance, etc), and developing a software package to run in closed-loop with visual feedback. I will comment on each of these items separately below. It would be great if the authors could more explicitly lay out the key contributions of this manuscript in the text.

The main objective of this work was to provide an open-source implementation of the real-time identification of motor units together with a user interface that allow researchers to easily process the data and display the firing activity of motor unit in the form of several visual feedback. We have explicitly laid out these key contributions in the introduction: “Having access to open-source tools that perform the real-time decoding of motor units would allow an increasing number of researchers to improve and expand the range of these applications.’

Related to the above, much of the validation of the algorithms in this manuscript has a "trust me" feel. The authors note that the Negro et al. algorithm has already been validated, so very few details or presentations of primary data showing the algorithm's performance are shown. Similarly, the efficacy of the decomposition approach is evaluated using manual editing of the sorting output as a reference, which is a subjective process, and users would greatly benefit from explicit guidance. There are very few details of manual editing shown in this manuscript (I believe the authors reference the Hug et al. 2021 paper for these details), and little discussion of the core challenges and variability of that process, even though it seems to be a critical step in the proposed workflow. So this is very hard to evaluate and would be challenging for readers to replicate.

To address the reviewer’s comment, we added a validation step using synthetic EMG data (P.10; L.235).

‘Validation of the algorithm

We first validated the accuracy of the algorithm using synthetic EMG signals generated with an anatomical model entailing a cylindrical muscle volume with parallel fibres [see Farina et al. (29), Konstantin et al. (36) for a full description of the model]. In this model, subcutaneous and skin layers separate the muscle from a grid of 65 surface electrodes (5 columns, 13 rows), while an intramuscular array of electrodes is directly inserted in the muscle under the grid with an angle of 30 degrees. 150 motor units were distributed within the cross section of the muscle. Recruitment thresholds, firing rate/excitatory drive relations, and twitch parameters were assigned to each motor unit using the same procedure as Fuglevand et al. (37). During each simulation, a proportional-integral-derivative controller adjusted the level of excitatory drive to minimise the error between a predefined target of force and the force generated by the active motor units.

Figure 3A displays the raster plots of the active motor units during simulated trapezoidal isometric contractions with plateaus of force set at 10%, 20%, and 30% MVC. A sinusoidal isometric contraction ranging between 15 and 25% MVC at a frequency of 0.5 Hz was also simulated. We identified on average 10 ± 1 and 12 ± 2 motor units with surface and intramuscular arrays, respectively (Figure 3A). During the offline decomposition, the rate of agreement between the identified discharge times and the ground truth, that is, the simulated discharge times, reached 100.0 ± 0.0% for intramuscular EMG signals and 99.2 ± 1.8% for surface EMG signals (Figure 3B). The offline estimation of motor unit filters was therefore highly accurate, independently of the level of force or the pattern of the isometric contraction.

Motor unit filters estimated during a baseline contraction at 20% MVC were then applied in real-time on signals simulated during a contraction with a different pattern (sinusoidal; Figure 3C). The rates of agreement between the online decomposition and the ground truth reached 96.3 ± 4.6% and 98.4 ± 2.3% for surface and intramuscular EMG signals, respectively. Finally, we tested whether the accuracy of the online decomposition changed when the level of force decreased or increased by 10% MVC when compared to the calibration performed at 20% MVC (Figure 3D). The rate of agreement remained high when applying the motor unit filters on signals recorded at 10% MVC: 99.8 ± 0.2% (surface EMG) and 99.5 ± 0.3% (intramuscular EMG). It is worth noting that only 3 out of 10 motor units identified from surface EMG at 20% MVC were active at 10% MVC, while 8 out of 12 motor units identified from intramuscular EMG were active at 10 % MVC. This shows how the decomposition of EMG signals tends to identify the last recruited motor units, which often innervate a larger number of fibres than the early recruited motor units (38). On the contrary, the application of motor unit filters on signals simulated at 30% MVC led to a decrease in the rate of agreement, with values of 88.6 ± 14.0% (surface EMG) and

80.3 ± 19.2% (intramuscular EMG). This decrease in accuracy did not impact all the motor units, with 5 motor units keeping a rate of agreement above 95% in both signals. For the other motor units, we observed a decrease in precision, which estimates the ratio of true discharge times over the total number of identified discharge times. This was caused by the recruitment of two motor units sharing a similar space within the muscle, which resulted in a merge in the same pulse train (Figure 3D).’

In addition, we added a new paragraph in the Method section to describe the manual editing process (P.26; L.658).

‘There is a consensus among experts that automatic decomposition should be followed by visual inspection and manual editing (55). Manual editing involves the following steps: (i) removing spikes that result in erroneous firing rates (outliers), (ii) adding discharge times thar are clearly distinguishable from the noise, (iii) recalculating the separation vector, (iv) reapplying the separation vector on the EMG signals (either a selected window or the entire signal), and (v) repeating this procedure until no outliers are present and all clearly distinguishable spikes have been selected. Importantly, the manual editing of potentially missed or falsely identified discharge times should not be accepted before the application of the updated motor unit separation vector, thereby generating a new pulse train. Manual edits should be accepted only if the silhouette value improves following this operation or remains well above the preestablished threshold. A more extensive description of the manual editing of motor unit pulse trains can be found in (32). Even though some of the aforementioned steps involve subjective decision-making, evidence suggests that manual editing after EMG decomposition with blind source separation approaches remains highly reliable across operators (33). Specifically, the median rates of agreement calculated for 126 motor units over eight operators with various experience in manual editing was 99.6%. All raw and processed data have been made available on a public data repository so that they can be used for training new operators (10.6084/m9.figshare.13695937).’

I found the User Guide in the Github package to be easy to follow. Importantly, it seems heavily tied to the specific hardware (Quattrocento). I understand it may be difficult to make the full software package work with different hardware, but it seems important to at least make an offline analysis of recorded data possible for this package to be useful more broadly.

The software was updated to perform real-time decomposition with signals recorded from the Quattrocento and the Open Ephys GUI, which is compatible with Intan and Open Ephys acquisition boards. I-Spin has also been adapted by TMSi to perform real-time decomposition with their devices (https://info.tmsi.com/blog/ispin-saga-real-time-motor-unit-decomposition-tool).

Moreover, the manual editing panel of the software can now import any files from these devices and allow users to reformat data in mat files to perform offline analyses.

While this may be a powerful platform, it is also very possible that without more details and careful guidance for users on potential pitfalls, many non-experts in sorting could use this as a platform for somewhat sloppy science.

We fully agree with the reviewer that real-time EMG decomposition - with a different approach here than spike sorting - may yield unreliable results if not applied properly. As outlined in the introduction of our initial manuscript, assessing the accuracy and limitations of real-time decomposition was a primary motivation for this study. Specifically, we compared accuracy between contraction intensities, muscles, and electrode types (see Results section).

We also demonstrated that manual editing of the decomposition outputs should be done after the training phase to improve the motor unit filters, thereby improving the accuracy of real-time decomposition. We also outlined the importance to never blindly accept the result of the decomposition without visual inspection and manual editing. (P8; L214)

‘These results show how manual editing can improve the accuracy of spike detection from the motor unit pulse trains. Moreover, a SIL value around 0.9 can be used as a threshold to automatically remove the motor unit pulse trains with a poor quality a priori. Thus, these two steps were performed in the all the subsequent analyses. Importantly, it is worth noting that the motor unit pulse train must always be visually inspected after the session to check for errors of the automatic identification of discharge times.’

We have also included more detailed information about the manual editing process (see above).

The authors mention that data is included with the Github software package. I could not find any included data, or instructions on how to run the software offline on example data.

This link to the data on figshare was added in the GitHub.

Given the centrality of the real-time visual feedback to their system, the authors should show some examples of the actual display etc. so readers can understand what the system in action actually looks like (I believe there is no presentation of the actual system in the manuscript, just in the User Guide). Similarly, it would be helpful to have a schematic figure outlining the full workflow that a user goes through when using this system.

A figure of the workflow is present in the user manual. Additionally, we now display traces of visual feedback in figure 5 and we added videos of the software during each of the visual feedback in supplemental materials.

The authors note all data was collected with male subjects because more motor units can be decomposed from male subjects relative to females. But what is the long-term outlook for the field if studies avoid female subjects because their motor units may be harder to decompose? This should at least be discussed - it is an important challenge for the field to solve, and it is unacceptable if new methods just avoid this problem and are only tested on male subjects.

This point was rightly raised by each of the three reviewers. To solve this, we added data collected on four females, and discussed future developments to make the decomposition of surface EMG equally performant for everyone (P.20; L.480).

‘An important consideration regarding the implementation of offline or real-time surface EMG decomposition is the difference between individuals, with an overall lower yield in number of identified motor units in females (here: 9 ± 12) than in males (here: 30 ± 13). Typically, the number of identified motor units from surface EMG is twice as low in females than males (32, 49, 50). The cause for this difference remains unclear. It may be related to variations in properties of the tissues separating the motor units from the recording electrodes, or to differences in the morphological and physiological properties of muscle fibres, as well as to the innervation ratios of motor units. These sex-related differences have so far only been supported by data extracted from animal experiments (51). However, the recent developments of simulation frameworks capable of generating highly realistic EMG signals for anthropometrically diverse populations may help understanding the impact of sex-related differences in humans (52). Specifically, these simulations can account for diverse anatomical (e.g. muscle volume and architecture, thickness of subcutaneous tissues) and physiological characteristics (e.g. innervation ratio, number of motor units, fibre cross sectional area, fibre conduction velocity, contribution of rate coding vs. spatial recruitment). Generating such dataset could help identifying the primary factors affecting EMG decomposition performance, ultimately enabling the refinement of algorithms and/or surface electrode design.’

Specific comments on the core contributions of this paper:C1. Development of an open-source implementation of the Negro algorithmThis seems an important contribution and useful for the community. There are very few figures showing any primary data, the efficacy of sorting, raw traces showing the waveforms that are identified, cluster shapes, etc. I realize the high-level algorithm has been outlined elsewhere, but the implementation in this package, and its efficacy, is a core component of the system and the claims being made in this paper. Much more presentation of data is needed to evaluate this.

It is worth noting that the approach used here is based on blind source separation, which is different than spike-sorting algorithms as it relies on the statistical properties of the spike trains (their sparseness) rather than the profiles of the action potentials. In short, we optimise separation vectors that are applied onto the whitened signal to generate a sparse motor unit pulse train. The discharge times are then directly estimated from the high peaks of this pulse train (Section 1 of the results; overview of the approach).

We are thus displaying motor unit pulse trains in three figures with the automatically detected discharge times, with cases of successful separation in figure 1 and merged motor units in the same pulse train in figures 3 and 4.

We also validated the algorithm with synthetic EMG to provide objective data on the accuracy of the algorithm. These results are shown in the section ‘Validation of the algorithm’ and displayed in figure 3.

Similarly, more information on the offline manual editing process (e.g. showing before/after examples with primary data) would be important to gain confidence in the method. The current paper shows application to both surface EMG and intramuscular EMG, but I could not find IM EMG examples in the Hug paper (apologies if I missed them). Surface and IM data are very, very different, so one would imagine the considerations when working with them should also be different.

In response to another comment from the reviewer, we have included more detailed information about the manual editing process (see above). As stated above, the decomposition approach used in our software differs from a spike sorting approach. Therefore, even though intramuscular and surface EMG signals are different, the decomposition and manual editing process is the same.

All descriptions of math/algorithms are presented in text, without any actual math, variable definitions, etc. This presentation makes it difficult to understand what is done. I would strongly recommend writing out equations and defining variables where possible.More details on how the level of sparseness is controlled during optimization would be helpful.And how this sparseness penalty is weighed against other optimization costs.

A mathematical description of the model has been added in the methods (P25; L620)

‘Mathematical modelling of the recorded spike trains.

The spike train of a motor neuron recorded over time 𝑡 ∈ [0, 𝑇] can be described as the result of a convolution between a delta function (d) representing the firing times (j), and finite impulse responses (h) representing action potentials of duration L: ∑l=0L−1h(l)∑rδ(t−φr−l). In practice, the nature of h and the duration L depend on the type of recordings. For electrophysiological measurements, h characterises the local electrical field generated by the spike and conducted through the surrounding tissues.

As the recorded volume of tissue comprises many active neurons, each recording can be considered as a convolutive mixture of multiple sources, and the previous equation can be expressed in the form of a matrix to also consider all the electrodes of an array: given ∑rδ(k−φr);X(t)=∑l=0L−1H(l)S(k−l)+N(t), where X(t)=[x1(t),x2(t),…,xm(t)]T is a matrix of m electrophysiological signals, S(t)=[s1(t),s2(t),…,sn(t)]T is a matrix of n motor neurons’ spike trains, and 𝐻(𝑙) is a m by n matrix containing the lth sample of action potentials from n neurons and m signals. In this situation, we can reformulate the model as an instantaneous mixture of an extended set of sources, that is, the motor neurons’ spike trains and their delayed versions. This allows us to simply write the previous equation as a multiplication of matrices, in which each source is delayed L times, L being the duration of the impulse response h. This model can be inverted for neural decoding with source-separation approaches.’

The rest of the decomposition approach was rewritten to make it clearer for the reader:

‘The monopolar EMG signals collected during the baseline contractions were extended with an extension factor of 1000/m (21), where m is the number of channels free of any noise or artifact. The signals were then demeaned and whitened. A contrast function was iteratively applied to estimate a separation vector that maximised the level of sparseness of the motor unit pulse train (Figure 1B). This loop stopped when the variation of the separation vector between two successive iterations reaches a predefined lower bound. After the application of a peak detection algorithm, the motor unit pulse train contained high peaks (i.e., the spikes from the identified motor unit) and low peaks from other motor units and noise. High peaks were separated from low peaks and noise using K-mean classification with two classes (Figure 1B). The peaks from the class with the highest centroid were considered as spikes of the identified motor unit. A second algorithm refined the estimation of the discharge times by iteratively recalculating the separation vector and repeating the steps with peak detection and K-mean classification until the coefficient of variation of the inter-spike intervals was minimised. The accuracy of each estimated spike train was assessed by computing the silhouette (SIL) value between the two classes of peaks identified with K-mean classification (24). When the SIL exceeded a predetermined threshold, the motor unit filter was saved for the real-time decomposition, together with the centroids of the ‘spikes’ and ‘noise’ classes (Figure 2A).’

Overall the paper is not very rigorous about the accuracy of motor unit identification. For example, the authors note that SIL of 0.9 is generally used for offline evaluation (why is this acceptable?), but it was lowered to 0.8 for particular muscles in this study. But overall, it is unclear how sorting accuracy/inaccuracy affects performance in the target applications of this work.

In the section mentioned by the reviewer, we aimed to show how this metric can help to automatically select motor units that are likely to have a higher accuracy of spike detections as the peaks of their pulse train are easily separable from the noise.

We reformulated the conclusion of this section to make it clearer (P8; L214):

‘These results show how manual editing can improve the accuracy of spike detection from the motor unit pulse trains. Moreover, a SIL value around 0.9 can be used as a threshold to automatically remove the motor unit pulse trains with a poor quality a priori. Thus, these two steps were performed in the all the subsequent analyses. Importantly, it is worth noting that the motor unit pulse train must always be visually inspected after the session to check for errors of the automatic identification of discharge times.’

C2. For real-time experiments, variability/jitter is important to characterize. Fig. 4 seems to be presenting mean computational times, etc, but no presentation of variability is shown. It would be helpful to depict data distributions somehow, rather than just mean values.

The variability in computational time was added to this section (P.28; L.730):

‘The standard deviation of computational times across windows reached 5.4 ± 4.0 ms (raster plot), 4.0 ± 3.2 ms (smoothed firing rate), and 2.8 ± 2.5 ms (quadrant)’

The computational time minimally varied between the successive windows, except when the labels of the x-axis were updated in real-time with scrolling feedback. It was overall always well below the duration of the window.

Computational time for each iteration of the algorithm in one participant. The top panels display the continuous computation time through the recording, while the bottom panels display the distribution of computational times. The dash line represents the duration of a window of EMG signals.

There is some description about the difference between units identified during baseline contractions, and how they might be misidentified during online contractions ("Accuracy of the real-time identification..."). This should be described in more detail.

We added an additional section in the results to clarify the concept of motor unit filters, and the reapplication of motor unit filters on signals in real-time. We highlighted how each motor unit must have a unique spatio-temporal signature to be accurately identified by our algorithms, in opposition to merged motor units sharing the same spatio-temporal features. This section shows how motor units accurately identified during baseline contractions can be misidentified during online contractions (P12; L295).

‘Application of motor unit filters in experimental data

We then asked eight participants (4 males and 4 females) to perform trapezoidal isometric contractions with plateaus of force set at 10% and 20% MVC during which surface EMG signals were recorded from the TA with 256 electrodes separated by 4 mm. The aim of this experiment was to confirm the results of the simulation; specifically, to test the accuracy of the online decomposition when the level of force was below, equal to, or above the level of force produced during the baseline contraction used to estimate the motor unit filters (Figure 4). We assessed the accuracy of the motor unit spike trains identified in real time using their manually edited version as reference. 144 motor units were identified at both 10 and 20% MVC. When the test signals were recorded at the same level of force as the baseline contraction, we obtained rates of agreement of 95.6 ± 6.8% (10% MVC) and 93.9 ± 5.9% (20% MVC). The sensitivity reached 95.9 ± 6.7% (10% MVC) and 94.4 ± 5.6% (20% MVC), and the precision reached 99.6 ± 1.3% (10% MVC) and 99.4 ± 1.9% (20% MVC).

When the filters identified at 20% MVC were applied on signals recorded at a lower level of force (10% MVC), the rates of agreement decreased to 87.9 ± 16.2%. The sensitivity also decreased to 88.0 ± 16.2%, but the precision remained high (99.4 ± 4.3). Thus, the decrease in accuracy was mostly caused by missed discharge times rather than the false identification of artifacts or spikes from other motor units.

When the filters identified at 10% MVC were applied to signals recorded at a higher level of force, the rates of agreement decreased to 83.3 ± 13.5%. The sensitivity decreased to 90.7 ± 8.1%, and the precision also decreased to 90.9 ± 12.6%. This result confirms what was observed with synthetic EMG, that is motor units recruited between 10 and 20% MVC can substantially disrupt the accuracy of the decomposition in real-time, as highlighted in Figure 4 (lower panel). Importantly, this situation does not happen for all the motor units, as suggested by the distribution of the values in Figure 4.’

Fig. 6: Given that a key challenge in sorting should be that collisions occur during large contractions, much more primary data should be presented/visualized to show how the accuracy of sorting changes during larger contractions in online experiments.

As indicated above, the decomposition approach implemented in our software is not based on spikesorting, so it does not require to separate overlapping profiles of action potentials (see Methods).

Fig.7: In presenting the accuracy of biofeedback, it is very hard to gain any intuition for performance by just looking at RMSE values. Showing the online decoded and edited trajectories would help readers understand the magnitude of errors.

We updated the figure to display examples of visual feedback before and after manual editing.

**Reviewer #3 (Public Review):**
In this manuscript, Rossato and colleagues present a method for real-time decoding of EMG into putative single motor units. Their manuscript details a variety of decision points in their code and data collection pipeline that led to a final result of recording on the order of ~10 putative motor units per muscle in human males. Overall, the manuscript is highly restricted in its potential utility but may be of interest to aficionados. For those outside the field of human or nonhuman primate EMG, these methods will be of limited interest.

We thank the reviewer for his/her throughout evaluation of our manuscript. We recognise that this tool/resource will immediately benefit groups working with humans or nonhuman primate models. However, the recent development of intramuscular thin films with various designs adapted to rodents and smaller animals could expand the range of future users (Chung et al., 2023, Elife). Nonetheless, decoding motor units in humans could be useful for many fields, e.g. in the domains of movement restoration and augmentation. The following paragraph has been added in the introduction section to highlight the importance of real-time decoding of motor unit activity (P3; L67):

‘The activity of motor neuron – in the form of spike trains – represents the neural code of movement to muscles. Decoding this firing activity in real-time during various behaviours can thus substantially enhance our understanding of movement control (2-5). Real-time decoding is also essential for interfacing with external devices (6) or virtual limbs (7) when activity is present at the periphery of the nervous system. For example, individuals with a spinal cord injury can control a virtual hand with the residual firing activity of the motor units in their forearm (7). Furthermore, sampling the activity of motor units receiving a substantial portion of independent synaptic inputs may pave the way for movement augmentation – specifically, extending a person’s movement repertoire through the increase of controllable degrees of freedom (8). In this way, Formento et al. (3) showed that individuals can intuitively learn to independently control motor units within the same muscle using visual cues. Having access to open-source tools that perform the real-time decoding of motor units would allow an increasing number of researchers to improve and expand the range of these applications.’

Notes(1) Artificial data should be used with this method to provide ground truth performance evaluations. Without it, the study assumptions are unchallenged and could be seriously flawed.

A new section on the validation of the algorithm has been added. We verified the accuracy of the algorithm by comparing the series of identified discharge times with the ground truth, i.e., the simulated discharge times. (P10; L235)

‘Validation of the algorithm

We first validated the accuracy of the algorithm using synthetic EMG signals generated with an anatomical model entailing a cylindrical muscle volume with parallel fibres [see Farina et al. (29), Konstantin et al. (36) for a full description of the model]. In this model, subcutaneous and skin layers separate the muscle from a grid of 65 surface electrodes (5 columns, 13 rows), while an intramuscular array of electrodes is directly inserted in the muscle under the grid with an angle of 30 degrees. 150 motor units were distributed within the cross section of the muscle. Recruitment thresholds, firing rate/excitatory drive relations, and twitch parameters were assigned to each motor unit using the same procedure as Fuglevand et al. (37). During each simulation, a proportional-integral-derivative controller adjusted the level of excitatory drive to minimise the error between a predefined target of force and the force generated by the active motor units.

Figure 3A displays the raster plots of the active motor units during simulated trapezoidal isometric contractions with plateaus of force set at 10%, 20%, and 30% MVC. A sinusoidal isometric contraction ranging between 15 and 25% MVC at a frequency of 0.5 Hz was also simulated. We identified on average 10 ± 1 and 12 ± 2 motor units with surface and intramuscular arrays, respectively (Figure 3A). During the offline decomposition, the rate of agreement between the identified discharge times and the ground truth, that is, the simulated discharge times, reached 100.0 ± 0.0% for intramuscular EMG signals and 99.2 ± 1.8% for surface EMG signals (Figure 3B). The offline estimation of motor unit filters was therefore highly accurate, independently of the level of force or the pattern of the isometric contraction.

Motor unit filters estimated during a baseline contraction at 20% MVC were then applied in real-time on signals simulated during a contraction with a different pattern (sinusoidal; Figure 3C). The rates of agreement between the online decomposition and the ground truth reached 96.3 ± 4.6% and 98.4 ± 2.3% for surface and intramuscular EMG signals, respectively. Finally, we tested whether the accuracy of the online decomposition changed when the level of force decreased or increased by 10% MVC when compared to the calibration performed at 20% MVC (Figure 3D). The rate of agreement remained high when applying the motor unit filters on signals recorded at 10% MVC: 99.8 ± 0.2% (surface EMG) and 99.5 ± 0.3% (intramuscular EMG). It is worth noting that only 3 out of 10 motor units identified from surface EMG at 20% MVC were active at 10% MVC, while 8 out of 12 motor units identified from intramuscular EMG were active at 10 % MVC. This shows how the decomposition of EMG signals tends to identify the last recruited motor units, which often innervate a larger number of fibres than the early recruited motor units (38). On the contrary, the application of motor unit filters on signals simulated at 30% MVC led to a decrease in the rate of agreement, with values of 88.6 ± 14.0% (surface EMG) and 80.3 ± 19.2% (intramuscular EMG). This decrease in accuracy did not impact all the motor units, with 5 motor units keeping a rate of agreement above 95% in both signals. For the other motor units, we observed a decrease in precision, which estimates the ratio of true discharge times over the total number of identified discharge times. This was caused by the recruitment of two motor units sharing a similar space within the muscle, which resulted in a merge in the same pulse train (Figure 3D).’

(2) From the point of view of a motor control neuroscientist studying movement in animals other than humans or non-human primates, the title was misleadingly hopeful. The use case presented in this study requires human participants to perform isometric contractions, facilitating spatially redundant recordings across the muscle for the algorithm to work. It is unclear whether these methods will be of utility to use cases under more physiological conditions (ie. dynamic movement).

We modified the title to read: “I-Spin live: An open-source software based on blind-source separation for real-time decoding of motor unit activity in humans”.

(3) The text states that "EMG signals recorded with an array of electrodes can be considered and instantaneous mixture of the original motor unit spike trains and their delayed versions." While this may be a true statement, it is not a complete statement, since motor units at distal sites may be shared, not shared, or novel. It was not clear to me whether the diversity of these scenarios would affect the performance of the software or introduce artifacts. In other words, if at site 1 you can pick up the bulk signal of units 1,2,3,4; at site two you pick up the signals of units 2,3,4,5 and site three you pick up the signal of units 3,4,5,6, what does the algorithm assume is happening and what does it report and why?

This section has been rewritten to clarify this point. The EMG signal represents indeed the sum of the active motor units within the recorded muscle volume. Put in other words, it is possible that deep motor units or motor units with innervated fibres far away from the grid were not in this recorded muscle volume, and thus non-identifiable. Another necessary condition to ensure the identifiability of the motor unit is its unique spatio-temporal signature within the signal. It means that two motor units close to each other within the muscle volume will be merged by the model. This point was clarified in the results during the validation and the application of filters on experimental data.

(P5; L115)

‘An EMG signal represents the sum of trains of action potentials from all the active motor units within the recorded muscle volume (Figure 1A). During stationary conditions, e.g., isometric contractions, the train of motor unit action potentials can be modelled as the convolution of series of discrete delta functions, representing the discharge times, and motor unit action potentials that have a consistent shape across time. When EMG signals are recorded with an array of electrodes, the shape of the recorded potential of each motor unit differs across electrodes. This is due to (1) the varying conduction velocity of action potentials among the muscle fibres, and (2) the location/depth of the muscle fibres that belong to each motor unit relatively to the electrodes, which impact the low pass filtering effect of the tissue on the recorded potential. Increasing the number and density of recording electrodes increases the likelihood that each motor unit will have a unique motor unit action potential profile (shape), i.e., a temporal and spatial profile that differs from all the other active motor unit within the recorded volume (16, 29). The uniqueness of motor unit action potential profiles is necessary for the blind source separation to accurately estimate the motor unit discharge times. Conversely, the spike trains of two motor units with similar action potential profiles will be merged by the model.

Our software uses a fast independent component analysis (fastICA) to retrieve motor unit spike trains from the EMG signals. For this, it iteratively optimises a separation vector (i.e., the motor unit filter) for each motor unit [Figure 1B; (24-26)]. (24-26). The projection of the EMG signals on this separation vector generates a sparse motor unit pulse train, with most of its samples close to zero and a smaller number of samples significantly greater than zero (Figure 1B). The discharge times are estimated from this motor unit pulse train using a peak detection function and a k-mean classification with two classes to separate the high peaks (spikes) from the low peaks (noise and other motor units). During the decomposition in real-time, short segments of EMG signals are projected on the saved separation vectors, and the peaks are classified as discharge times if they are closer to the centroid of the class ‘spikes’ than to the centroid of the class ‘noise’ (Figure 1C). The algorithm used to identify motor units discharge activity is based on that proposed by Negro et al. (24) and Barsakcioglu et al. (26).’

(4) I could not fully appreciate the performance gap solved by the current methods. What was not achievable before that is now achievable? The 125 ms speed of deconvolution? What was achievable before? Intro text around ln 85 states that 'most of the current implementations of this approach rely on offline processing, which restricts its ability to be used..." but no reference is provided here about what the non 'most' of can achieve.(8) The authors might try to add text to be more circumspect about the contributions of this method. I would recommend emphasizing the conceptual advances over the specifics of the performance of the algorithm since processor speed and implementation of the ideas in a faster environment (Matlab can be slow) will change those outcomes in a trivial way. Yet, much of the results section is very focused on these metrics.

The main contribution of this work submitted to the section ‘Tools and Resource’ of Elife is to provide a user interface that enables researchers to decompose EMG signals recorded with multichannel systems into motor unit activities, to perform this process in real-time, and to translate it into visual feedback. The user interface is fully open source and does not require coding experience. If necessary, the users can inspect the commented code and even modify it for their own experimental setup. The toolbox is now compatible with various acquisition boards, which can expand its use to novel surface and intramuscular arrays of electrodes.

(5) Relatedly, it would have been nice to see a proof of concept using real-time feedback for some kind of biofeedback signal. If that is the objective here, why not show us this? I found the actual readout metrics of performance rather esoteric. They may be of interest to very close experts so I will defer to them for input.

We agree with the reviewer. Videos were added to the supplemental materials to show the different forms of feedback, together with a case scenario where the participant try to separate the activity of two motor units from the same muscle.

(6) I was disappointed to see that only male participants are used because of some vague statement that 'it is widely known in the field' that more motor units can be resolved in males, without thorough referencing. It seems that the objective of the algorithm is the speed of analysis, not the number of units, which makes the elimination of female participants not justified.

The reviewer is right and that was corrected in the new version of the manuscript. We first performed additional experiments in both males and females focused on the accuracy of the approach, and further discussed the differences in yield between men and women in the discussion together with research perspectives to solve this issue.

Results (P12; L296):

‘We then asked eight participants (4 males and 4 females) to perform trapezoidal isometric contractions with plateaus of force set at 10% and 20% MVC during which surface EMG signals were recorded from the TA with 256 electrodes separated by 4 mm. The aim of this experiment was to confirm the results of the simulation; specifically, to test the accuracy of the online decomposition when the level of force was below, equal to, or above the level of force produced during the baseline contraction used to estimate the motor unit filters (Figure 4). We assessed the accuracy of the motor unit spike trains identified in real time using their manually edited version as reference. 144 motor units were identified at both 10 and 20% MVC. When the test signals were recorded at the same level of force as the baseline contraction, we obtained rates of agreement of 95.6 ± 6.8% (10% MVC) and 93.9 ± 5.9% (20% MVC). The sensitivity reached 95.9 ± 6.7% (10% MVC) and 94.4 ± 5.6% (20% MVC), and the precision reached 99.6 ± 1.3% (10% MVC) and 99.4 ± 1.9% (20% MVC).

When the filters identified at 20% MVC were applied on signals recorded at a lower level of force (10% MVC), the rates of agreement decreased to 87.9 ± 16.2%. The sensitivity also decreased to 88.0 ± 16.2%, but the precision remained high (99.4 ± 4.3). Thus, the decrease in accuracy was mostly caused by missed discharge times rather than the false identification of artifacts or spikes from other motor units. When the filters identified at 10% MVC were applied to signals recorded at a higher level of force, the rates of agreement decreased to 83.3 ± 13.5%. The sensitivity decreased to 90.7 ± 8.1%, and the precision also decreased to 90.9 ± 12.6%. This result confirms what was observed with synthetic EMG, that is motor units recruited between 10 and 20% MVC can substantially disrupt the accuracy of the decomposition in real-time, as highlighted in Figure 4 (lower panel). Importantly, this situation does not happen for all the motor units, as suggested by the distribution of the values in Figure 4.’

Discussion (P20; L480):

“An important consideration regarding the implementation of offline or real-time surface EMG decomposition is the difference between individuals, with an overall lower yield in number of identified motor units in females (here: 9 ± 12) than in males (here: 30 ± 13). Typically, the number of identified motor units from surface EMG is twice as low in females than males (32, 49, 50). The cause for this difference remains unclear. It may be related to variations in properties of the tissues separating the motor units from the recording electrodes, or to differences in the morphological and physiological properties of muscle fibres, as well as to the innervation ratios of motor units. These sex-related differences have so far only been supported by data extracted from animal experiments (51). However, the recent developments of simulation frameworks capable of generating highly realistic EMG signals for anthropometrically diverse populations may help understanding the impact of sex-related differences in humans (52). Specifically, these simulations can account for diverse anatomical (e.g. muscle volume and architecture, thickness of subcutaneous tissues) and physiological characteristics (e.g. innervation ratio, number of motor units, fibre cross sectional area, fibre conduction velocity, contribution of rate coding vs. spatial recruitment). Generating such dataset could help identifying the primary factors affecting EMG decomposition performance, ultimately enabling the refinement of algorithms and/or surface electrode design.”

(7) Human curation is often used in spike sorting, but the description of criteria used in this step or how the human curation choices are documented is missing.

To address the reviewer’s comment, we added a new paragraph in the Method section to describe the manual editing process: (P26; L657)

“There is a consensus among experts that automatic decomposition should be followed by visual inspection and manual editing (55). Manual editing involves the following steps: (i) removing spikes that result in erroneous firing rates (outliers), (ii) adding discharge times thar are clearly distinguishable from the noise, (iii) recalculating the separation vector, (iv) reapplying the separation vector on the EMG signals (either a selected window or the entire signal), and (v) repeating this procedure until no outliers are present and all clearly distinguishable spikes have been selected. Importantly, the manual editing of potentially missed or falsely identified discharge times should not be accepted before the application of the updated motor unit separation vector, thereby generating a new pulse train. Manual edits should be accepted only if the silhouette value improves following this operation or remains well above the preestablished threshold. A more extensive description of the manual editing of motor unit pulse trains can be found in (32). Even though some of the aforementioned steps involve subjective decision-making, evidence suggests that manual editing after EMG decomposition with blind source separation approaches remains highly reliable across operators (33). Specifically, the median rates of agreement calculated for 126 motor units over eight operators with various experience in manual editing was 99.6%. All raw and processed data have been made available on a public data repository so that they can be used for training new operators (10.6084/m9.figshare.13695937).”

MinorLn 115, "inversing" is not a word. "inverse" is not a verb

Changed as suggested

Ln 186, typo, bioadhesive

Changed as suggested

MVC should be defined on first use. It is currently defined on 3rd use or so.The term rate is used in a variety of places without units. Eg line 465 but not limited to that

Changed as suggested

**Recommendations for the authors:**

**Reviewer #1 (Recommendations For The Authors):**
Two minor comments: Para 125: it is not clear what is meant by "spatial distribution" of recording electrodes.

‘Density’ was used instead of ‘spatial distribution’ to now read:

‘Increasing the number and density of recording electrodes increases the likelihood that each motor unit will have a unique motor unit action potential profile (shape), i.e., a temporal and spatial profile that differs from all the other active motor unit within the recorded volume (16, 29).’

Para 545: perhaps a bit more explanation about why low spatial overlap is better would be appropriate.

We added a section in the results showing how motor units with similar spatial signatures are merged by our model, leading to a lower precision. We therefore changed this sentence to now read:

‘Therefore, the likelihood of having spatially overlapping motor unit action potentials - and thus merged motor units - is lower, which explains why the rate of agreement of motor units identified from intramuscular arrays of electrodes is much higher than grids of surface electrodes (12, 13).’

**Reviewer #2 (Recommendations For The Authors):**
The authors mention that data is included with the Github software package. I could not find any included data, or instructions on how to run the software offline on example data. (Apologies if I missed this - it would be helpful to make it more prominent)

The link to the data on figshare was added in the GitHub, as well as data samples to run the algorithm offline and test manual editing.

Minor comments:Not sure what is meant by "boundary capabilities of online decomposition"

This was removed to only discuss the accuracy of online decomposition.

CoV for ISIs is not formally defined or justified.

This was added to the caption of figure 2:

‘The CoV of ISI estimates the regularity of spiking for each motor unit, an expected behaviour during isometric contractions at consistent levels of force.’

Fig. 4: slope units should be ms/motor unit, perhaps?

Changed as suggested.

In some places, the manuscript uses "edition" to describe the editing process. I am not familiar with this usage, "editing" may be more common.

Editing is now used through the entire manuscript.

**Reviewer #3 (Recommendations For The Authors):**
I would recommend that the authors revise their manuscript to conform to eLife formatting guidelines, including moving the methods to the end of the manuscript. This change may entail substantial editing since many ideas are presented in order from the beginning of the methods. While this suggestion may seem superficial, the success of the new publishing model might benefit from general uniformity in manuscript style.

We changed and edited the draft to follow the classic format of Elife papers.